# Fast and Accurate Causal Parallel Decoding using Jacobi Forcing

**Lanxiang Hu** [* 1]  **Siqi Kou** [* 2]  **Yichao Fu** [1]  **Samyam Rajbhandari** [3]
**Tajana Rosing** [1]  **Yuxiong He** [3]  **Zhijie Deng** [2]  **Hao Zhang** [1]

## Abstract

Multi-token generation has emerged as a promising paradigm for accelerating language model inference, with the diffusion Large Language Models (dLLMs) as the most notable approach recently. Popular dLLMs like SDAR and Fast-dLLM v2 are post-trained on pre-trained AR models to minimize training cost while maintaining high performance. However, there exists a fundamental *pretrain-to-posttrain mismatch* – the masked data distribution and bidirectional attention in post-training deviates significantly from the real data distribution and causal attention for pretraining. As a result, the post-trained dLLMs usually suffer from limited speedup or substantially degraded performance. To address this, we introduce *Jacobi Forcing* to bypass the dLLM formulation, directly post-training a *causal* multi-token predictor from an AR LLM. In particular, we force the model to learn to leap along its own parallel token generation trajectories based on Jacobi Decoding, and introduce an elaborate progressive distillation paradigm. The trained models achieve $3.8\times$ wall-clock speedup on coding and math benchmarks with minimal loss in performance. Based on the trajectory characteristics of the model, we further introduce multi-block decoding with rejection recycling, which enables up to $4.6\times$ higher token acceptance count per iteration and $4.0\times$ wall-clock speedup, effectively trading additional compute for lower inference latency.

## 1. Introduction

Modern large language models (LLMs), such as GPT-5 (OpenAI, 2025), Gemini-2.5 (DeepMind, 2025), and Kimi-K2 (Team et al., 2025), excel at complex and interactive agentic tasks. Yet, autoregressive (AR) decoding generates tokens sequentially, limiting parallelism and leading to high latency. To address this, recent work explores predicting multiple future tokens natively in transformer-based models without relying on auxiliary draft models (Gloeckle et al., 2024; Liu et al., 2024a). A popular approach is diffusion-based language models (dLLMs), which relax left-to-right generation by modeling the entire sequence jointly and decoding via full-sequence denoising (Nisonoff et al., 2024; Schiff et al., 2024; Inception Labs, 2025). Despite the promise to enable highly parallelizable computation, open pretrained dLLMs (Ye et al., 2025; Zhu et al., 2025; Nie et al., 2025a) underperform AR models in generation quality.

Recent efforts adapt high-quality AR LLMs into dLLMs for parallel decoding to preserve generation quality (JetAstra, 2025; Wu et al., 2025b). Concretely, they perform block-wise perturbations to the data and post-train AR LLMs with block-wise bidirectional attention and mask prediction objective. However, this adaptation delivers limited speedup under quality constraints. This is due to a significant *pretrain-to-posttrain mismatch* – the masked data distribution and bidirectional attention in post-training deviates significantly from the real data distribution and causal attention for pretraining. As a result, SDAR (Cheng et al., 2025) suffers substantial quality drops when large block sizes (*e.g.*, 64 or 128) are adopted, and AR-adapted dLLMs are costly to train (as shown in Figure 1). The inability to use large block sizes also leads to the underutilization of modern AI accelerators with abundant FLOPs.

To address this, we introduce *Jacobi Forcing* to bypass the dLLM formulation, directly post-training a fast and accurate causal parallel decoder from an AR LLM. It post-trains AR LLMs on their own parallel generation trajectories from Jacobi Decoding. Jacobi Decoding is a widely adopted parallel decoding technique for AR LLMs (Song et al., 2021; Santilli et al., 2023). It first randomly initializes a block of $n$ tokens and feeds it to the AR LLMs to iteratively update it, and eventually, the block converges to the same $n$ tokens generated by AR decoding, forming a trajectory between the randomly initialized point and the converged point. The full sequence is generated block by

---

[1]University of California, San Diego [2]Shanghai Jiaotong University [3]Snowflake. Correspondence to: Hao Zhang <haozhang@ucsd.edu>, Zhijie Deng <zhijied@sjtu.edu.cn>.

*Proceedings of the 43rd International Conference on Machine Learning*, Seoul, South Korea. PMLR 306, 2026. Copyright 2026 by the author(s).

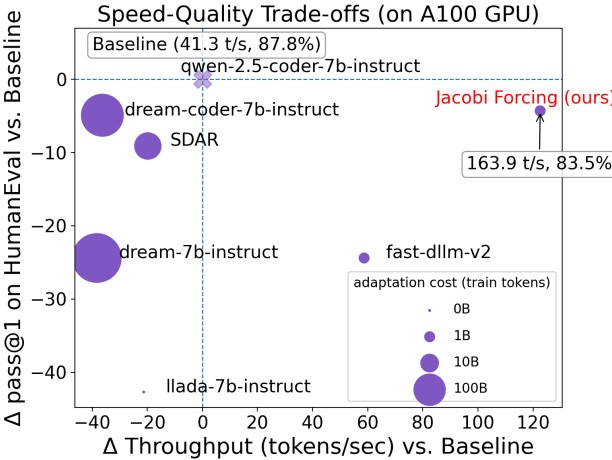

*Figure 1.* Speedup-quality tradeoff with training adaptation costs for JF-LLM and baseline parallel decoding models.

block. Prior works including CLLM (Kou et al., 2024) and CEED-VLA (Song et al., 2025a) design a consistency loss to map any point along the trajectory to the converged point, which in turn teaches AR LLMs to predict multiple correct tokens in one iteration simultaneously.

*Jacobi Forcing* (JF) improves over the baselines by introducing a noise-aware causal attention that teaches the model to predict the converged point within each block conditioned on previous unconverged blocks. This addresses a limitation of existing works including AR-adapted dLLMs – as block size increases, the number of tokens correctly decoded per iteration remains essentially constant. We show that it enables more useful future tokens to emerge in the trailing tail and even in subsequent blocks, creating opportunity for higher token acceptance rate as block size scales. Furthermore, *Jacobi Forcing* repeats this distillation procedure for the trained model and involves more noisy data with a larger block size for progressive distillation.

We observe JF-LLM has a stronger capability of generating correct future tokens conditioning on noisy context, consistent with our training objective. To better utilize this characteristic, we design a rejection-recycling and multi-block decoding algorithm for further inference optimization. Rejection recycling reuses high-quality consecutive tokens discarded from past decoding iterations to generate candidate token sequences, enabling the decoding of more accurate tokens via verifying multiple branches in a single iteration. Multi-block decoding maintains and refines multiple blocks simultaneously, where correct tokens are decoded in subsequent blocks even when preceding blocks remain unconverged for further speedup.

Experiments show JF-LLM can serve as very efficient parallel decoders with up to $3.8\times$ improvement in generation speed across coding and math benchmarks. It also effec-

tively generates higher quality draft n-grams from future tokens within each block, as observed in Section 5. Using rejection-recycling and multi-block decoding makes use of future $n$-grams and further boost speedup to around $4\times$, and token-per-forward (TPF) to $4.6\times$ as block size scales.

In summary, key contributions of this paper includes:

- We introduce *Jacobi Forcing* to train AR models as fast parallel decoders, JF-LLM, with up to $3.8\times$ generation speedup with vanilla greedy Jacobi decoding.
- We empirically observe and quantitatively verify JF-LLM has both higher TPF count and a useful $n$-gram count in comparison with baseline models.
- We propose rejection-recycling and multi-block decoding to make use of higher quality draft $n$-grams from future tokens within each block. Apply them to JF-LLM boost generation speed to $4.0\times$, and TPF to up to $4.6\times$ across various benchmarks as block size scales. On B200 GPU with our vLLM implementation, the techniques enable a JF-LLM decoding throughput at 865 tokens/s.

## 2. Related Work

**Discrete Text Diffusion.** dLLMs represent a new paradigm that challenges traditional autoregressive (AR) modeling by replacing left-to-right causality with iterative denoising, enabling parallel multi-token generation (Li et al., 2024a; Nisonoff et al., 2024; Schiff et al., 2024). Closed-source dLLMs (e.g., Gemini Diffusion (Google DeepMind, 2025; Inception Labs, 2025; Song et al., 2025b)) show huge throughput improvement while maintaining competitive code and text quality, underscoring better accelerator utilization. On the open-source side, community dLLMs with released code and weights delivered strong throughput and controllability via parallel iterative denoising, yet remaining less efficient than autoregressive decoding (Ye et al., 2025; Zhu et al., 2025; Nie et al., 2025a; JetAstra, 2025; Gong et al., 2025). Recent efforts (Arriola et al., 2025; Wu et al., 2025c; Liu et al., 2025) further push the efficiency and scalability of dLLMs.

**Jacobi Decoding.** Jacobi decoding reframes AR generation as a parallel fixed-point update over all positions, with convergence linked to greedy AR, and has been instantiated using Jacobi (Gauss-Seidel) iterations (Song et al., 2021; Santilli et al., 2023). Building on this, follow-ups either refine the decoding procedure or train models as parallel decoders to exploit parallel: CLLMs (Kou et al., 2024) fine-tune LLMs with consistency distillation to predict multiple correct tokens per iteration and speed convergence; CEED-VLA (Song et al., 2025a) brings the similar idea to robotics. Other strands adapt Jacobi to new regimes, including FastCoT (Zhang et al., 2023) for reasoning with

parallel CoT updates, Speculative Jacobi Decoding (Teng et al., 2024) for sampling in AR Text-to-Image, and MSN, TR-Jacobi (Wang et al., 2024) that injects denoising training and a retrieval-augmented Jacobi strategy.

**Speculative Decoding**. Speculative decoding (SD) speeds up AR generation by letting a lightweight drafter propose several future tokens and having the target model verify them in one pass (Leviathan et al., 2022; Chen et al., 2023). It preserves the target model's distribution while reducing latency. Subsequent work improves proposal quality and verification efficiency: online speculative decoding (OSD) (Liu et al., 2024b) adapts draft models to user query distributions via continual distillation, substantially improving token acceptance rate and reducing inference latency. VocabTrim (Goel et al., 2025) prunes draft model's output vocab to reduce computation while maintaining effective speculative proposals. Medusa (Cai et al., 2024) adds multi-head drafters to the base LM to produce verifiable token blocks; EAGLE, EAGLE-2 (Li et al., 2024b;c) reuse target features for feature-level drafting, and EAGLE-3 (Li et al., 2025) scales this idea with multi-layer fusion. Lookahead Decoding (Fu et al., 2024), PLD (Saxena, 2023; Somasundaram et al., 2024), and REST (He et al., 2023) dispense with a separate drafter, instead synthesizing speculative candidates directly from context or future tokens. The self-speculative decoding paradigm shares a close connection with the Jacobi decoding adopted in this work.

## 3. Preliminary

This section reviews the basics of Jacobi decoding and consistency distillation training to accelerate Jacobi decoding of AR models.

### 3.1. Jacobi Decoding

Given a prompt $\boldsymbol{x}$ and a pre-trained LLM $p_\theta(\cdot|\boldsymbol{x})$ parametrized by $\theta$, the standard AR decoding under the greedy strategy produces a response sequentially as follows:

$$y_i = \arg\max_y p_\theta(y \mid \boldsymbol{y}_{<i}, \boldsymbol{x}), \quad \text{for } i = 1, \ldots, n, \quad (1)$$

where $\boldsymbol{y}_{<i} = \{y_1, \ldots, y_{i-1}\}$. This process requires $n$ forward passes of the LLM to generate $n$ tokens $\boldsymbol{y}_{\leq n}$. The inherently sequential nature of AR decoding limits practical efficiency when generating long sequences. Jacobi decoding (Song et al., 2021; Santilli et al., 2023) addresses this bottleneck by reformulating token generation as solving a system of nonlinear equations:

$$f(y_i, \boldsymbol{y}_{<i}, \boldsymbol{x}) = 0, \quad \text{for } i = 1, \ldots, n, \quad (2)$$

where $f(y_i, \boldsymbol{y}_{<i}, \boldsymbol{x}) := y_i - \arg\max_y p_\theta(y|\boldsymbol{y}_{<i}, \boldsymbol{x})$. This system can be solved in parallel using Jacobi fixed-point

iteration (ort, 2000). Starting from a randomly initialized $n$-token sequence $\boldsymbol{y}^{(0)} = \{y_1^{(0)}, \ldots, y_n^{(0)}\}$, the update at each iteration $j$ is:

$$\begin{cases} y_1^{(j+1)} &= \arg\max_y p_\theta(y|\boldsymbol{x}) \\ y_2^{(j+1)} &= \arg\max_y p_\theta(y|\boldsymbol{y}_1^{(j)}, \boldsymbol{x}) \\ &\vdots \\ y_n^{(j+1)} &= \arg\max_y p_\theta(y|\boldsymbol{y}_{<n}^{(j)}, \boldsymbol{x}). \end{cases} \quad (3)$$

Notably, for LLM, the above $n$ maximization problems can be solved in parallel by using a causal attention mask, i.e., only one forward pass of the LLM is required to obtain $\boldsymbol{y}^{(j+1)}$ based on $\boldsymbol{y}^{(j)}$. The iteration exits at some $k$ such that $\boldsymbol{y}^{(k)} = \boldsymbol{y}^{(k-1)}$ and we define $\boldsymbol{y}^* := \boldsymbol{y}^{(k)}$ as the fixed point. Let $\mathcal{J} := \{\boldsymbol{y}^{(0)}, \ldots, \boldsymbol{y}^{(k)}\}$ denote the Jacobi trajectory. It can be proven that $\boldsymbol{y}^*$ is identical to AR decoding under greedy strategy (Song et al., 2021).

To generate a long response $\boldsymbol{l}$ of length $L \gg n$, Jacobi decoding is applied sequentially over blocks of size $n$ until the `<eos>` token appears in a fixed point. Let $\boldsymbol{y}_{B_i}^*$ denote the fixed point obtained for the $i$-th block. The full output $\boldsymbol{l}$ is then constructed by concatenating fixed points from consecutive blocks:

$$\boldsymbol{l} = [\boldsymbol{y}_{B_1}^*, \ldots, \boldsymbol{y}_{B_N}^*], \quad (4)$$

where $N = \lceil \frac{L}{n} \rceil$ denotes the number of blocks generated before termination.

### 3.2. Consistency Distillation

Despite the promise, Jacobi decoding achieves little speedup over standard AR decoding (Santilli et al., 2023; Fu et al., 2024), as it rarely predicts more than one correct[1] token within one fixed-point iteration. To address this, recent works such as CLLMs (Kou et al., 2024) propose consistency distillation, a training approach designed to accelerate convergence to the fixed point from arbitrary states on a Jacobi trajectory. The key idea is to introduce a consistency loss that encourages an LLM $p_\theta(\cdot|\boldsymbol{x})$ to predict multiple tokens simultaneously:

$$\mathcal{L}_c = \mathbb{E}_{i \sim \mathcal{U}\{1,\ldots,N\}, \boldsymbol{y}_{B_i} \sim \mathcal{J}_i} \Big[ D_{\text{KL}}\big( p_{\theta^-}\big(\boldsymbol{y}_{B_i}^*|\boldsymbol{x}, \boldsymbol{y}_{B_{<i}}^*\big) \\ \| \, p_\theta\big(\boldsymbol{y}_{B_i}|\boldsymbol{x}, \boldsymbol{y}_{B_{<i}}^*\big)\big) \Big], \quad (5)$$

where $\theta^- = \text{stopgrad}(\theta)$ and $D_{\text{KL}}$ denotes the KL divergence aggregated across the $n$ tokens in a block. Here, $i \sim \mathcal{U}\{1, \ldots, N\}$ denotes sampling a block index uniformly at random, and $\boldsymbol{y}_{B_i} \sim \mathcal{J}_i$ denotes randomly sampling from the Jacobi trajectory of the $i$-th block.

---

[1]By correctness, we mean alignment with the AR decoding result under a greedy sampling strategy.

CLLMs build upon this idea by first collecting Jacobi trajectories, obtained by running Jacobi decoding with $p_\theta$ on a set of prompts. The model is then trained with a joint objective that combines the consistency loss in Eq. 5 with the standard AR loss, achieving up to a $2\times$ speedup over AR decoding while maintaining quality. Similar training objectives have also been adopted for inference acceleration in other domains, such as action prediction in VLA models (Song et al., 2025a).

## 4. Methodology

In this section, we first discuss the training challenges of consistency distillation with larger block sizes $n$, and then present *Jacobi Forcing*, a progressive consistency distillation method designed to mitigate this bottleneck, and denote LLMs trained under this paradigm as JF-LLM. Furthermore, by observing JF-LLM's trajectories under vanilla Jacobi decoding, we introduce rejection-recycling and multi-block decoding strategies to improve its efficiency.

### 4.1. Jacobi Forcing

**Progressive Noise Schedule.** In Jacobi decoding, we maintain strict causality within each block, where each token is updated in accordance with Eq. 3. Consider the $i$-th block $\boldsymbol{y}_{B_i}^{(j)}$ of size $n$ is been decoded at some iteration step $j$. Assume the first $c - 1$ tokens have been accepted, and we denote $y_f$ as the future token as shown in Eq. 6.

$$
y_f = \arg\max_y \, p\big(y \mid \boldsymbol{x}_c, \, \boldsymbol{y}'_{c:f-1}\big),
$$
$$
\text{for } f = c + 1, \ldots, n, \tag{6}
$$

where $\boldsymbol{x}_c = [\boldsymbol{x}, \boldsymbol{y}_{<c}]$ is the clean context, $\boldsymbol{y}'_{c:f-1}$ is the noisy[2] context. While the training objective in Eq. 5 is designed to optimize correct token prediction in this setting, it's observed from (Kou et al., 2024) that predicting $y_f$ is hard when it's conditioned on a long noisy context $\boldsymbol{y}'_{c:f-1}$ under large block sizes (e.g., $n = 256$).

To address this challenge, we instead split a large block into smaller blocks (e.g., $n = 16$) with noise ratios determined by a predefined schedule $\{t_1, \ldots, t_N\}$. Each $t_i$ denotes the fraction of noisy tokens in a block. The noise schedule follows a cyclic strategy with window size $w$, where the noise ratio linearly increases from 0 to 1 within each window, i.e.,

$$
W = \left\{0, \frac{1}{w}, \ldots, \frac{w-1}{w}\right\}, \quad t_i = W[j], \quad j = i \bmod w. \tag{7}
$$

[2] By noisy, we refer to tokens in the non-converged point along the Jacobi trajectory that that differ from those in the fixed point at the same positions.

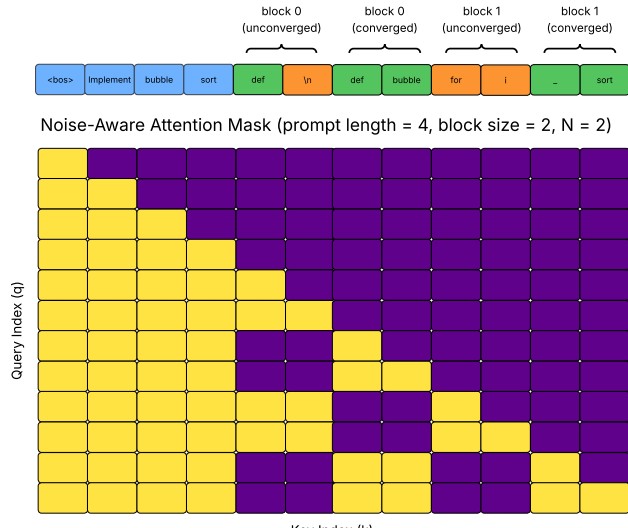

*Figure 2.* Illustration of oisy-context conditioned causal mask implementations. Both allow logits from clean blocks and noisy blocks to be generated with a single forward pass to compute the progressive consistency loss and AR loss in Eq. 9.

This progressive schedule ensures that each block retains a partially clean context, thereby shortening noisy tokens dependencies. In particular, it reduces the longest span of consecutive noisy inputs for any prediction from $O(nN)$ assuming $t_i = 1$ for all blocks using a random schedule to $O(\lceil tn \rceil)$ using a progressive schedule, which facilitates learning. Empirically, we find this progressive schedule to be more effective than a purely random noise schedule (Table 4).

**Progressive Distillation Loss.** Let $\boldsymbol{y}_{b_i}^{t_i}$ denote the point along the $i$-th block Jacobi trajectory with several noisy tokens closest to $\lceil t_i n \rceil$. The training objective is to predict tokens correctly within each block, aggregating losses across blocks to reduce gradient variance and stabilize optimization. Accordingly, we introduce a new loss term, *progressive consistency loss*, which optimizes $p_\theta$ under the progressive noise schedule in Eq. 7:

$$
\mathcal{L}_{\mathrm{pc}} = \frac{1}{N} \sum_{i=1}^{N} D_{\mathrm{KL}}\Big(p_{\theta^-}\big(\cdot \mid \boldsymbol{x}, \boldsymbol{y}_{B_{<i}}^*\big)
$$
$$
\big\| \, p_\theta\big(\cdot \mid \boldsymbol{x}, \boldsymbol{y}_{B_{<i}}^t\big)\Big), \tag{8}
$$

**AR Loss.** Kou et al. (2024) notes that using only the consistency loss (Eq. 5) must be supplemented with an AR loss to maintain generation quality. Our preliminary experiments show that using only the consistency objective (Eq. 8) produces the same effect. This motivates our inclusion of a conventional AR loss term in the final training objective to safeguard output quality:

$$
\mathcal{L}(\theta) = \mathcal{L}_{\mathrm{pc}} + w\mathcal{L}_{\mathrm{AR}} \tag{9}
$$

where $w$ is a tunable weight that balances the two learning objectives.

**Noise-aware Causal Attention.** In CLLM, loss from each training step is computed based on KL divergence from one block instance in Eq. 5. This learning objective is to train correct token prediction in the setting where there is only a big block (Eq. 6). Moreover, in both Eq. 5 and Eq. 8, the loss term computation involves two forward passes using a conventional causal mask since each involves a distinction sequence. As a result, it requires $O(2N)$ forward passes to compute all loss terms in Eq. 8 and $O(N)$ backward passes to compute gradients, resulting in low training efficiency. We reduce the number of forward and backward passes from $O(N)$ to $O(1)$ by introducing a sequence packing technique and a block-wise sparse attention mask. We illustrate the sequence packing that interleaves $\boldsymbol{y}_{b_i}^{t_i}$ and $\boldsymbol{y}_{b_i}^*$ for the entire complete sequence in Figure 2 for $\mathcal{L}_{pc}$ computation.

**Progressive Distillation for Larger Block Sizes.** In training JF-LLM on trajectories from the original AR model, we find that speedup scales with training steps and saturates at large step counts, likely due to significant data distribution shifts from extensively trained models. To break this ceiling, we collect an additional round of Jacobi trajectories with *progressively larger block sizes* from the JF-LLM empowered with multi-token prediction capability and further train it on newly generated trajectories. This yields a further 20% speedup with only minor performance degradation. Detailed training configurations are in Section 5.1.

### 4.2. Inference Optimization

**Behavior of JF-LLM.** JF-LLM is trained to have a stronger capability of generating correct future tokens conditioning on noisy tokens. Qualitative analysis in Figure 4 illustrates that it indeed brings the quality improvement: fixed-point segments emerge within the noisy tokens of the unconverged point. Furthermore, these segments progressively extend (e.g., the number of red tokens increases from point 1 to point 2 in Figure 4), even under noisy context, consistent with our training patterns. In this section, we focus on how to translating this qualitative observation of draft quality improvement into qualitative speedup.

**Rejection Recycling.** Prior work has shown that n-grams produced during Jacobi iterations can be verified in parallel and reused in subsequent iterations (Fu et al., 2024). As illustrated in Figure 4, such n-gram sizes could be large in JF-LLM. If correctly verified, many tokens can be fast-forwarded in one iteration. In particular, we initialize a fixed-size n-gram pool constructed from noisy token sequences observed at unconverged points during Jacobi decoding. During decoding, if the pool contains an n-gram whose first token matches the last accepted token of the

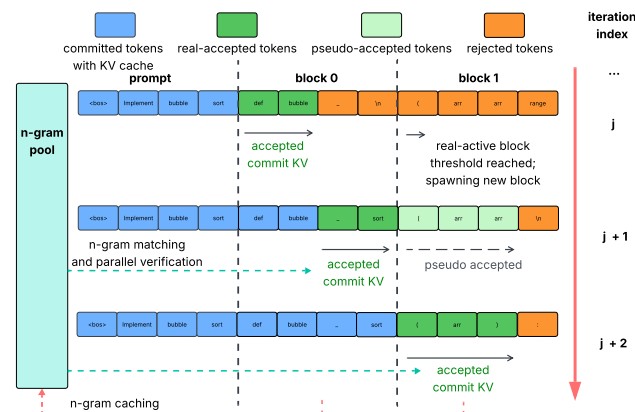

*Figure 3.* An example of multiblock decoding with rejection recycling at prompt length = 4, block size = 4, $r = 0.5$, $K = 2$.

current point, we extend this token by concatenating it with its subsequent tokens to form new candidates. These candidates are then verified in parallel by appending them along the batch dimension. At each iteration, we select the candidate that yields the largest number of newly accepted tokens. For instance, this strategy enables skipping from point 3 to point 5 in Figure 4, as the fixed-point segments in point 3 yield higher-quality candidates.

**Multi-block Decoding.** In addition to high-quality n-grams in the draft, we also observe the increasing number of stationary tokens, which are correctly predicted with preceding noisy tokens and remain unaltered through subsequent iterations. Together they yield higher quality drafts. To make use of the property, we introduce *multi-block decoding*, a new decoding paradigm that maintains and refines up to $K$ blocks simultaneously. It marks the block closest to the effective KV cache boundary as the *real-active* block and all the other $K - 1$ blocks as *pseudo-active* blocks. Only tokens within the real-active block are accepted and committed to KV cache. Tokens in pseudo-active blocks are only pseudo-accepted, conditioning on prior blocks; once converged, pseudo-active blocks will wait until they are promoted as the real active block, where all tokens will be verified again, but now with a higher-quality draft. A detailed description is provided in Algorithm 1 (with rejection recycling) in Appendix A and with an example in Figure 3. Note that both rejection recycling and multi-block decoding are lossless as they employ rejection sampling in the real-active block (Leviathan et al., 2022).

## 5. Experiments

### 5.1. Evaluation Settings

**Models and Datasets.** We evaluate JF-LLM across coding benchmark. For coding benchmarks, we train Qwen2.5-Coder-Insutrct (Hui et al., 2024) on OpenCodeInstruct (Ah-

*Figure 4.* Visualization of JF-LLM's trajectory under vanilla Jacobi decoding. The figure shows a partial segment of the trajectory. Blue tokens denote accepted tokens that match the fixed point at their positions. Black tokens denote unconverged noisy tokens, and we highlight them in red if more than three consecutive tokens match the fixed point regardless of position.

mad et al., 2025) and test on the HumanEval (Chen et al., 2021), MBPP (Austin et al., 2021). On OpenCodeInstruct, we curate question instances that come with generations that pass all unit tests, from where we use 450k prompts for trajectory generation and training. For mathematical tasks, we train Qwen2.5-Math-7B-Instruct (Yang et al., 2024) on the math split of Openthought2 (Guha et al., 2025) and test on GSM8K (Cobbe et al., 2021), and MATH (Hendrycks et al., 2021). On Openthought2, only mathematical prompts are considered, from where we apply the same settings for trajectory generation and training.

**Training Settings**. All training and inference are conducted on instances equipped with 8x NVIDIA A100-80GB GPUs and $8\times$ NVIDIA H200 GPUs. All models are trained with a learning rate of $10^{-6}$, a batch size of 4, and a max new sequence length of 2048. For JF-LLM, we adopt a linear progressive noise schedule, initial block size at 16, window size at 16, and train for 10k steps, and a second round of training with block size at 32, window size at 8, and train for another 10k steps. Ablation studies on parameter choices are presented in Section 5.3.

**Baselines.** Our main objective in this section is to compare performance and efficiency between various parallel decoding techniques and JF-LLM. Specifically, we compare JF-LLM with state-of-the-art (SOTA) dLLMs including LLaDA-7B (Nie et al., 2025b), Dream-7B (Ye et al., 2025), fast-dLLM-Dream-7B (Wu et al., 2025c) and D2F-Dream-7B (Wang et al., 2025). We also compare JF-LLM with existing AR-based parallel decoder, including vanilla Jacobi decoding (Santilli et al., 2023) and CLLM (Kou et al., 2024). For SD baselines, we compare against two recent techniques including EAGLE-3 (Li et al., 2025) and HASS (Zhang et al., 2025), which require training for additional draft head training but comes with higher acceptance rate. In Table 1 and Table 3, we compare with best models from authors. In Appendix B, we provide a further comparison against more speculative decoding baselines trained using Qwen2.5-7B and consistency-distilled baselines on a single B200 GPU.

### 5.2. Results

**Performance.** The performance metrics are the greedy generations' strict accuracy (pass@1) on HumanEval and

*Table 1.* Performance and efficiency on coding benchmarks, HumanEval and MBPP, grouped by decoding family. All AR-based methods adopt Qwen2.5-Coder-7B-Instruct. For JF-LLM, MR stands for employing the multi-block and rejection-recycling decoding algorithm introduced in Algorithm 1. DC stands for using bi-directional dual cache from fast-dLLM.

| Method | TPF ↑ | TPS ↑ | Speedup ↑ | Accuracy ↑ |
|---|---|---|---|---|
| **HumanEval** | | | | |
| *AR-based* | | | | |
| AR | 1.00 | 41.3 | 1.00× | 87.8 |
| EAGLE-3 | 6.60 | 127.2 | 3.07× | 87.8 |
| HASS | 4.82 | 95.2 | 2.31× | 87.8 |
| Jacobi | 1.03 | 39.9 | 0.97× | 87.8 |
| CLLM* | 2.68 | 103.3 | 2.50× | 87.8 |
| JF-LLM | 4.01 | 159.5 | 3.86× | 83.5 |
| JF-LLM (MR) | 4.09 | **163.9** | **3.97×** | 83.5 |
| *Diffusion-based* | | | | |
| LLaDA-Instruct | 1.00 | 2.8 | 0.07× | 36.0 |
| Dream-Base | 1.00 | 20.2 | 0.49× | 54.3 |
| Fast-dLLM (DC) | 1.80 | 60.0 | 1.45× | 53.0 |
| D2F | 2.50 | 73.2 | 1.77× | 54.3 |
| **MBPP** | | | | |
| *AR-based* | | | | |
| AR | 1.00 | 43.1 | 1.00× | 74.3 |
| EAGLE-3 | 3.46 | 90.5 | 2.10× | 74.3 |
| HASS | 3.20 | 79.7 | 1.85× | 74.3 |
| Jacobi | 1.01 | 42.4 | 0.98× | 74.3 |
| CLLM* | 2.10 | 80.1 | 1.94× | 71.4 |
| JF-LLM | 2.74 | 110.7 | 2.57× | 70.4 |
| JF-LLM (MR) | 2.84 | **113.0** | **2.62×** | 70.4 |
| *Diffusion-based* | | | | |
| LLaDA-Instruct | 1.00 | 0.9 | 0.02× | 39.0 |
| Dream-Base | 1.00 | 10.4 | 0.24× | 56.2 |
| Fast-dLLM (DC) | 1.90 | 73.2 | 1.70× | 51.0 |
| D2F | 2.30 | 105.0 | 2.44× | 55.2 |

MBPP. Table 1 compares JF-LLM with both dLLMs and Jacobi decoding baselines. On A100 GPUs, our results show that on both benchmarks, JF-LLM consistently achieves competitive accuracy with a much better speedup at the same parameter scale. In particular, for structured generations like Python coding, JF-LLM achieves $3.8\times$ speedup in comparison with the AR baseline, $53.3 \sim 7.4\times$ speedup comparing to dLLM baselines, and $2.0\times$ comparing to optimized dLLM baselines including Fast-dLLM and D2F with techniques like adding block-wise KV cache, bidirectional KV cache and pipelined parallel decoding. For speedup

*Table 2.* Speedup on HumanEval evaluated on B200 with the same settings as Table 1 and Table 3 at batch size = 1, with additional results from our vLLM implementation using CUDA graph and page attention kernel to reduce synchronization stalls.

| Framework | Method | TPF ↑ | TPS ↑ | Speedup ↑ |
|---|---|---|---|---|
| HuggingFace | AR | 1.00 | 83.0 | 1.00× |
| | Jacobi | 1.03 | 84.7 | 1.02× |
| | JF-LLM | 4.01 | 301.7 | 3.63× |
| | JF-LLM (MR) | 4.21 | **328.0** | **3.95×** |
| vLLM | AR | 1.00 | 221.3 | 1.00× |
| | Jacobi | 1.03 | 227.9 | 1.03× |
| | JF-LLM | 4.01 | 796.7 | 3.60× |
| | JF-LLM (MR) | 4.21 | **865.8** | **3.91×** |

evaluation, we run all evaluations with a block size of 128 and a block size of 64 for JF-LLM (MR) since MR takes extra FLOPs for multiblock decoding (block count = 2) and parallel verification. Detailed discussion on inference configuration choice is provided in Section 5.3.

*Table 3.* Performance and efficiency on math benchmarks, GSM8K and MATH, grouped by decoding family. All AR-based methods adopt Qwen2.5-Math-7B-Instruct.

| Method | TPF ↑ | TPS ↑ | Speedup ↑ | Solve Rate ↑ |
|---|---|---|---|---|
| **GSM8K** | | | | |
| *AR-based* | | | | |
| AR | 1.00 | 41.8 | 1.00× | 92.4 |
| EAGLE-3 | 6.89 | 132.0 | 3.16× | 92.4 |
| HASS | 6.68 | 127.6 | 3.05× | 92.4 |
| Jacobi | 1.05 | 42.2 | 1.02× | 92.4 |
| CLLM* | 2.25 | 86.8 | 2.08× | 92.2 |
| JF-LLM | 3.72 | 146.1 | 3.50× | 91.4 |
| JF-LLM (MR) | 4.04 | **154.9** | **3.71×** | 91.4 |
| *Diffusion-based* | | | | |
| LLaDA-Instruct | 1.00 | 7.2 | 0.17× | 77.4 |
| Dream-Base | 1.00 | 9.5 | 0.23× | 75.0 |
| Fast-dLLM (DC) | 2.10 | 49.8 | 1.19× | 75.0 |
| D2F | 3.10 | 91.2 | 2.18× | 77.6 |
| **MATH** | | | | |
| *AR-based* | | | | |
| AR | 1.00 | 41.3 | 1.00× | 77.0 |
| EAGLE-3 | 6.38 | 122.3 | 2.97× | 77.0 |
| HASS | 6.70 | 126.4 | 3.06× | 77.0 |
| Jacobi | 1.02 | 41.0 | 0.99× | 77.0 |
| CLLM* | 2.23 | 84.4 | 2.04× | 77.2 |
| JF-LLM | 3.82 | 150.7 | 3.65× | 77.4 |
| JF-LLM (MR) | 3.98 | **152.0** | **3.68×** | 77.4 |
| *Diffusion-based* | | | | |
| LLaDA-Instruct | 1.00 | 21.1 | 0.51× | 23.7 |
| Dream-Base | 1.00 | 9.9 | 0.24× | 35.8 |
| Fast-dLLM (DC) | 1.90 | 67.0 | 1.62× | 37.1 |
| D2F | 2.60 | 98.8 | 2.39× | 35.4 |

Moreover, we report the speedup and problem solve rate (test@1) on GSM8K and MATH in Table 3. Across both benchmarks, the Jacobi Forcing Model substantially outperforms the AR baseline with 3.7× speedup while preserving competitive accuracy. In the MATH benchmark,

*Table 4.* Ablation on different noise schedules using block size = 32, $t_{min}$ = 0.0 and $t_{max}$ = 1.0. Acc. = pass@1 accuracy (%) on HumanEval. The checkpoints are trained with Qwen2.5-Coder-7B-Instruct on 10k randomly sampled training instances. Reverse progressive is significantly worse than other schedules.

| Schedule | Window Size | Acc. | iter/token |
|---|---|---|---|
| Random | 8 | 82.9 | 0.53 |
| | 16 | 83.5 | 0.51 |
| | 32 | 83.5 | 0.53 |
| Linear Progressive | 8 | **84.7** | 0.48 |
| | 16 | 81.7 | 0.46 |
| | 32 | 84.1 | 0.49 |
| Reverse Progressive | 8 | < 80.0 | > 0.60 |
| | 16 | 82.9 | > 0.60 |
| | 32 | < 80.0 | > 0.60 |

JF-LLM delivers a 146.1 TPS while even slightly improving the solve rate from 77.0% to 77.4%, highlighting its ability to achieve both high efficiency and accuracy.

Since B200 comes with a better fast-forward count to TPS conversion rate with more compute on B200, we also present speedup comparison between different AR-based techniques with JF-LLM on B200 in Table 2, along results from our vLLM integration to support Jacobi decoding.

On B200, with the block size at 128 and verification size at 4 (rationale provided in Section 5.3), we apply multi-block decoding using JF-LLM and the results are presented in Figure 3. The running window method is an optimized variant of Jacobi decoding designed for settings where many tokens are accepted per iteration. It maintains a fixed-size active block by replenishing draft tokens to the original block size as accepted tokens are committed to the KV cache. The results demonstrate that multi-block decoding with rejection recycling consistently achieves the highest number of fast-forwarded tokens per iteration, particularly in the larger block-size regime as shown in Figure 5a.

## 5.3. Ablation Study

**Training Noise schedules.** We evaluate three types of noise schedules: random, linear progressive, and reverse progressive. In the random schedule, the noise step $t_i$ for each block is sampled uniformly as $t_i \sim \mathcal{U}(1, \ldots, N)$ during sequence packing in JF-LLM training. The linear progressive schedule follows Eq. 7, while the reverse progressive schedule applies a linearly decreasing noise ratio from 1 to 0 within each window. Results in Table 4 show that the linear progressive schedule significantly outperforms the other two when the window size is 8. Intuitively, with $n = 32$, this schedule corresponds to adding noise more aggressively across blocks within each window, roughly four additional noisy tokens per future block, until the final block where all tokens are noisy.

**Training Mask types.** We train JF-LLM on the objective in Eq. 8 with noise-conditioned mask implementation (Figure 2). For the noise-conditioned part of the query, each block is conditioned on progressively more noisy preceding blocks. Alternatively, we can set all previous blocks outside a window as clean context, while those inside the window remain noisy. We summarize results from this comparison in Table 5, where it shows full noise-conditioned mask is more effective in empowering JF-LLM while maintaining generation quality using the same data size. This effect is likely arise from a full noise-conditioned mask providing stronger learning signals for the model to denoise tokens within a single forward pass.

*Table 5.* Effects of applying noise-conditioned mask (NC) or noise-conditioned mask with intra-window clean context (NC-IC) for JF-LLM training on 100k examples, and evaluated on HumanEval with A100.

| Method | Speedup $\uparrow$ | Acc. |
|--------|--------------------|------|
| NC     | **3.6×**           | **82.3** |
| NC-IC  | 1.9×               | 82.3 |

**Inference FLOPs Utilization Analysis.** JF-LLM (MR) involves both multi-block decoding and rejection-recycling, where each technique consumes extra FLOPs for parallel drafting and parallel verification, respectively. To maximize hardware utilization, we experiment with how end-to-end decoding latency changes as the total number of decoded tokens changes. We use Jacobi decoding to run the experiments and the results are shown in Figure 5b. On H200 GPUs, Jacobi decoding with block sizes up to 64 shows no latency penalty and only minor degradation at 128, particularly in the high fast-forwarding regime. The result is consistent across accepted token counts fixed at $2, 3, 4, 5$, indicating that up to 126 tokens can be decoded in parallel with shared KV without significant latency overhead. We provide a more detailed analysis in Appendix D.

**Inference Configuration Search.** Beyond block size, the main tunable parameters for JF-LLM (MR) inference are verification size (entries verified in parallel with shared KV for rejection recycling), number of blocks, and initialization threshold. We observe that performance gains from additional blocks saturate at block size = 2 as later drafts degrade quickly. The initialization threshold, defined as the fraction of the first block completed before launching the next, can be optimized via grid search and shows consistently optimal performance at $r = 0.85$ for block size 64 across verification sizes 2 to 8. For maximum FLOPs utilization, we use block size = 64, verification size = 4, where wall-clock speedup remains stable until parallel decoding exceeds 256 tokens. More details on inference configuration search given the FLOPs budget can be found in Appendix E.

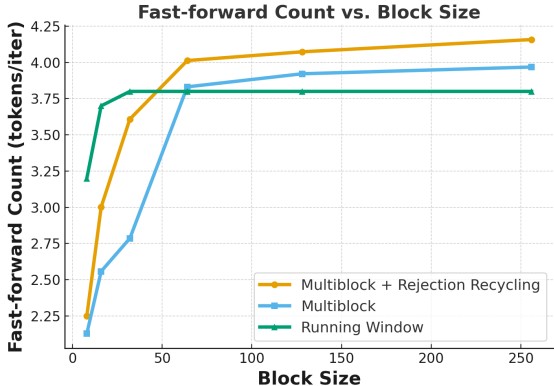

*(a)* TPF vs. block size on HumanEval using three decoding strategies. Running window refers to vanilla Jacobi decoding with fixed block sizes.

*(b)* TPS vs. (log-scaled) block size at different TPF, using Jacobi decoding at prompt length = 128, generation length = 256. Notice TPS starts to degrat at block size = 128.

*Figure 5.* Effect of block size choices on TPF and TPS on NVIDIA H200 GPU. We choose the maximum block size on hardware without sacrificing wall-clock speedup.

## 6. Conclusion

In this work, we propose a progressive distillation technique for training AR models as faster and more accurate parallel decoders compared to dLLMs. Unlike CLLM (Kou et al., 2024), which directly trains models to predict large blocks of tokens in parallel, our approach introduces a progressively more difficult learning objective. This is achieved through a progressive noise schedule, combined with a sequence packing strategy and a noise-aware causal mask, enabling parallel token prediction conditioned on noise. The model is further improved through iterative training, where trajectories are regenerated with progressively larger block sizes. The resulting model, JF-LLM, achieves a 3.8× speedup while largely preserving accuracy. Analysis of its generated trajectories shows that JF-LLM produces high-quality draft tokens toward the tail of sequences. In addition, we introduce rejection recycling and multi-block decoding, which together bring tokens accepted per iteration to 4.6× as high with nearly 4× speedup on HumanEval using on both A100 and B200 GPUs.

## Impact Statement

This work presents a challenge in machine learning and proposes a solution, the potential negative consequences are not apparent. While it is theoretically possible for any technique to be misused, the likelihood of such misuse occurring at the current stage is low.

## Acknowledgments

The work is supported by UCSD HDSI, Snowflake, NVIDIA, and a faculty research award from Google. We also gratefully acknowledge computing resources provided by Snowflake and NVIDIA through donation. Support was provided in part by the Shanghai Key Technology R&D Program "New Generation of Information Technology" (No. 25511103700), the NSF of China (Nos. 62306176, 92470118), the CCF-ALIMAMA TECH Kangaroo Fund (No. CCF-ALIMAMA OF 2025010), and Ant Group.

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

# A. Detailed Decoding Algorithm

---

**Algorithm 1** MULTIBLOCK DECODING + REJECTION RECYCLING

---

1: **Init:** Create a set of blocks $\{b\}$ with one *real–active* block $RA$: draft tokens $q_{RA}$ randomly initialized, accepted tokens $a_{RA} = \varnothing$ ; For all other blocks $b$, set $q_b = \varnothing$, $a_b = \varnothing$, and mark as *pseudo-active*.
2: Initialize candidate pool $\mathcal{N} = \emptyset$, spawn ratio $r$, threshold $s = \lceil rn \rceil$, block size $n$.
3: **while** iters $<$ max **do**
4:     **Assemble input $y$:** Concatenate $q_{RA}$, then for each pseudo-active $b$, append $a_b$ (no logits) and $q_b$ (collect logits). Resize cache to batch $y$.
5:     **Forward:** Run model $p_\theta(y)$ to produce logits.
6:     **for** each block $b$ with span $(start, L)$ **do**
7:         **Verification (with rejection-recycling):** Greedy prediction $g = \arg\max$ logits; accept longest matching prefix of $q_b$ using $g$ (or $g \cup \mathcal{N}$ if $b = RA$); update $a_b$.
8:         **if** $b = RA$ and EOS encountered in accepted region **then**
9:             **return** committed output.
10:         **end if**
11:         **Tail update:** If partial accept, set $q_b \leftarrow [\text{next}\|g_{\text{tail}}]$ (and if $b = RA$: push rejected tail to update $\mathcal{N}$ and $q_{RA}$); else $q_b \leftarrow \varnothing$.
12:     **end for**
13:     **Cache trim:** Delete false KV to committed length: prompt + verified $a_b$ (all accepted blocks) + $a_{RA}$.
14:     **Spawn:** If some block $b$ reaches $|a_b| \geq s$ and active $\{b\} < K$, clone and pad $q_{RA}$ to length $n$ and add as new pseudo-active block.
15:     **Promote:** If $|a_{RA}| \geq n$, choose a pseudo-active $b$ with $|a_b| > 0$, rebuild its draft to length $n$, mark as verified, set $RA \leftarrow b$.
16:     **Stop:** If all $|a_b| \geq n$ or EOS emitted by $RA$, break.
17: **end while**
18: **Finalize:** Concatenate output $=$ verified $a_b$ for all non-RA blocks, then $a_{RA}$; trim KV cache $\mathcal{C}$;
19: **Return:** (output, $\mathcal{C}$, iters)

---

We present the detailed algorithm for multi-block decoding and rejection sampling introduced in Section 4.2. Rejection recycling reuses high-quality consecutive tokens discarded in previous Jacobi iterations to construct candidate token sequences. Multi-block decoding jointly maintains and refines multiple blocks, allowing correct tokens in later blocks to be decoded even when earlier blocks remain unconverged, thereby further improving decoding throughput. These two techniques are orthogonal and can be seamlessly combined. As shown in Table 2, their combination yields an improvement of over 30 TPS compared to vanilla Jacobi decoding on a B200 GPU.

# B. Further Baseline Comparisons on B200

The main text focuses on comparisons between JF-LLM and diffusion-based parallel decoders, as well as AR-based parallel decoders, under a controlled setup where AR variants share the same backbone (Qwen2.5-Coder-7B-Instruct). This appendix extends the comparison to (i) distilled discrete diffusion models and (ii) state-of-the-art speculative decoding baselines.

**Distilled dLLM baselines.** A distilled dLLM baseline is useful for mapping JF-LLM against contemporary training techniques for discrete diffusion models. dParallel (Chen et al., 2025) performs trajectory-level consistency distillation on a discrete diffusion model to accelerate token sampling while aiming to preserve quality. We adopt the technique as the latest distilled dLLM baseline.

As shown in Table 6, on HumanEval, JF-LLM (MR) attains a noticeably stronger speed–quality profile than dParallel: JF-LLM (MR) achieves 29% higher accuracy and achieves more than 80% higher TPF and TPS. On GSM8K, JF-LLM improves accuracy by 8 absolute points with about 20% higher TPF and TPS (GSM8K numbers are omitted from the table below for brevity). These gaps indicate that, relative to latest consistency-distilled dLLM of comparable scale, JF-LLM occupies a more favorable point in the speed–quality trade-off space.

*Table 6.* Additional comparison on HumanEval across AR, speculative decoding, and dLLM-based methods. For the AR baseline and all Jacobi-decoding based methods, Qwen2.5-Coder-7B-Instruct is used as the backbone. Speedup is measured in TPS relative to the AR baseline on a single B200 GPU. Checkpoints marked in * refers to the strongest coding models released by the authors, which differ from the Qwen2.5-7B backbone.

| Family | Method | Acc. ↑ | TPF ↑ | TPS ↑ | Speedup ↑ |
|---|---|---|---|---|---|
| AR | AR (greedy) | 87.8 | 1.00 | 83.00 | 1.00× |
| dLLM | Fast-dLLM v2 | 63.4 | 1.00 | 83.29 | 1.00× |
| dLLM | SDAR | 78.7 | 2.36 | 31.46 | 0.38× |
| dLLM (distilled) | dParallel | 54.3 | 2.90 | 175.15 | 2.11× |
| AR + SpecDec | EAGLE-3 | 87.8 | 6.60 | 259.41 | 3.13× |
| AR + SpecDec | EAGLE-3* | 68.9* | 6.38 | 246.10 | 2.97× |
| AR + SpecDec | HASS | 87.8 | 4.82 | 193.50 | 2.33× |
| AR + SpecDec | HASS* | 61.6* | 5.53 | 280.29 | 3.37× |
| AR + Jacobi | Jacobi | 87.8 | 1.05 | 84.70 | 1.02× |
| AR + Jacobi | CLLM | 87.8 | 2.68 | 207.40 | 2.50× |
| AR + Jacobi | JF-LLM | 83.5 | 4.01 | 301.65 | 3.63× |
| AR + Jacobi | JF-LLM (MR) | 83.5 | 4.21 | **327.96** | **3.95×** |

**Speculative decoding and recent dLLM baselines.** Speculative decoding (SD) forms widely used family of AR acceleration methods. To place JF-LLM among such approaches, this appendix includes comparisons against two recent SD methods, EAGLE-3 (Li et al., 2025) and HASS (Zhang et al., 2025), which represent stronger baselines than earlier methods such as Medusa and Medusa-2. For these two methods, we compare JF-LLM against both the best checkpoints released by the authors and the models we trained using the same data as JF-LLM.

The comparison in Table 6 also includes two recent dLLM baselines, Fast-dLLM v2 (Wu et al., 2025a) and SDAR (Cheng et al., 2025), in addition to the community dLLM and D2F variants discussed in the main text. Fast-dLLM v2 improves blockwise diffusion efficiency via enhanced scheduling and caching, while SDAR introduces a synergistic diffusion–autoregressive paradigm for scalable sequence generation.

## C. Mapping Noise Schedule to Training Sequence for Progressive Consistency Distillation

We elaborate the process of mapping the noise schedule to arrive at the training sequence in Figure 2.

For each training sample, let the target model's complete generation of length $L$ be $y$. Given a training-time block size $n$ and a noise schedule $W$ (e.g., the linear progressive schedule in Eq. 2), we partition $y$ into $N = \lceil L/n \rceil$ blocks of size $n$. The schedule $W$ is applied over a window of $w$ blocks, yielding noise ratios $t_i$ defined in Eq. 7. For each block, we select the point along its Jacobi trajectory whose fraction of unconverged tokens (number of unconverged tokens/$n$) is closest to $t_i$, and use that point to form the corresponding noisy block. A full illustration is shown in Figure 6.

A complete training sequence contains both noisy and clean blocks. Clean blocks are the original partitions of $y$, while noisy blocks are constructed as above. We interleave each noisy block with its corresponding clean block so that a single forward pass, together with the custom attention mask in Figure 4, produces teacher logits on clean blocks for the AR loss and student logits on noisy blocks for the consistency loss. Under the progressive noise schedule, the longest consecutive noisy span within any block is $O(\lceil tn \rceil)$, which is much smaller than the naive $O(nN)$ worst case where every token in every block is noisy.

## D. Understanding TPF and FLOPs Trade-off

To estimate how many tokens can be decoded in parallel before hitting the hardware roofline, we profile generation-only latency as a function of the total number of simultaneously decoded tokens (horizontal axis in Figure 7), sweeping several block sizes $n_{\text{token\_seq\_len}}$. On H200 and B200 (left and middle panels), the curves for $n_{\text{token\_seq\_len}} \in \{16, 32, 64, 128\}$ are essentially flat as we increase the parallel token count up to $\approx 256$ tokens, and only start to grow noticeably when we push beyond that to 512 tokens. This plateau followed by an approximately linear region is the empirical roofline: up to $\sim 256$ batched tokens the GPU has spare FLOPs and KV bandwidth, so extra tokens are almost "free," whereas beyond that point the device becomes compute- or memory-bound and latency scales roughly linearly.

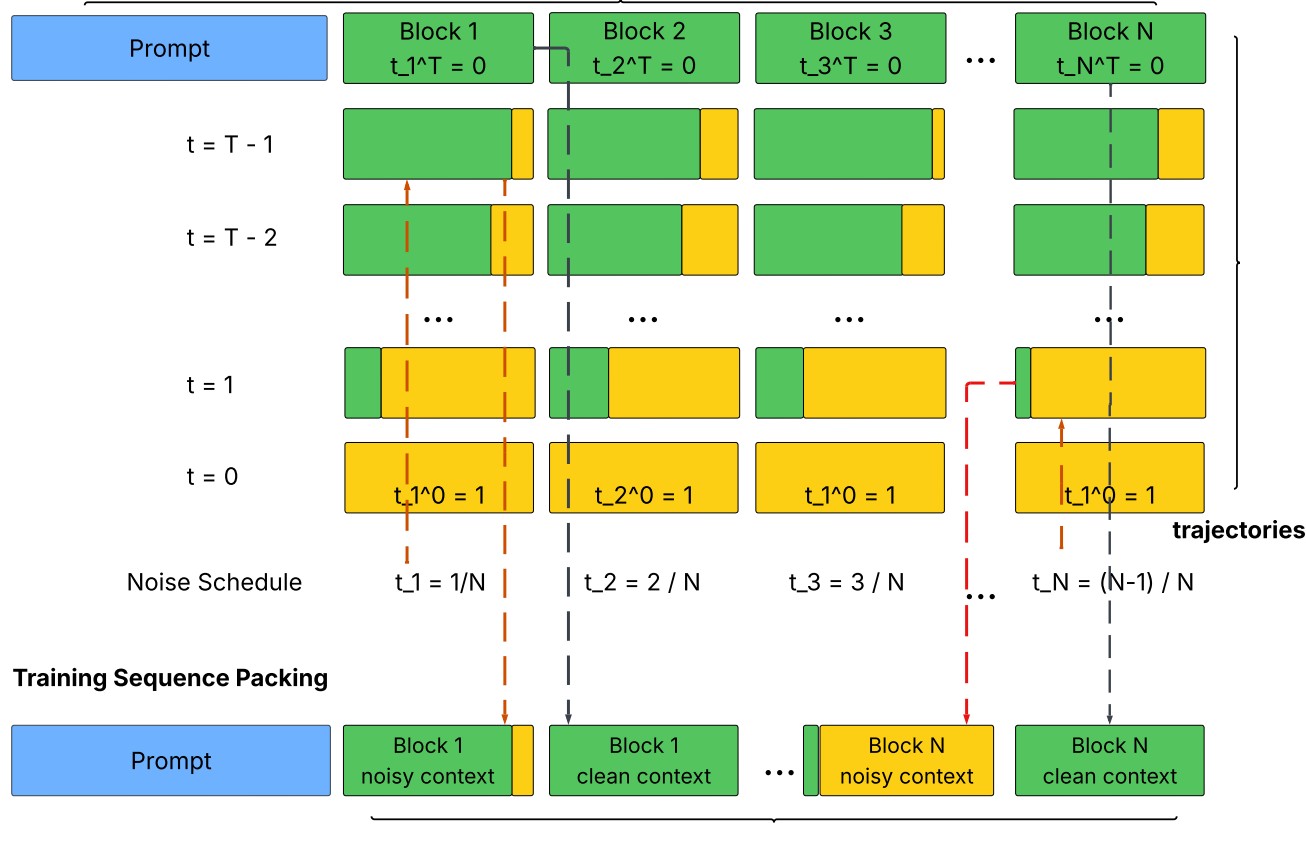

*Figure 6.* Illustration of the progressive noise schedule and training sequence packing. For each block $i$ over a total of $T_i$ decoding steps, we select the trajectory step whose fraction of unconverged tokens matches the scheduled noise ratio $t_i$ to form a noisy block (dashed red line), and pair it with the corresponding clean block (dashed dark line). The packed training sequence at the bottom interleaves all noisy and clean blocks, yielding $2N$ blocks so that a single forward pass can compute both AR and consistency losses.

On A100 (right panel of Figure 7), the plateau is shorter: generation time is nearly constant up to $\sim 128$ parallel tokens, but increases steeply once we go beyond 128 and approaches linear scaling by 256 tokens. Taken together, these measurements suggest operating near the "knee" of each roofline, which corresponds to $\approx 128$ parallel tokens on A100 and $\approx 256$ parallel tokens on H200/B200. This motivates our final configuration: block size $64$ with verification size $4$ on H200 and B200 ($64 \times 4 = 256$ tokens), which maximizes FLOPs utilization without hurting wall-clock performance.

These roofline measurements imply a FLOPs budget on each GPU: once the parallel token count approaches the hardware knee, additional tokens incur an almost linear increase in cost. Consequently, there is an explicit TPF–FLOPs tradeoff: configurations with larger blocks and more aggressive parallelism achieve higher TPF, but the extra FLOPs consumption can saturate the hardware and even degrade wall-clock latency.

## E. Inference Configuration Search

Because of this TPF-FLOPs trade-off, choosing an inference configuration is no longer a matter of simply maximizing block size or verification depth: **the configuration must respect the FLOPs budget implied by the roofline of the target GPU**. Once $K = 2$ and $r = 0.85$ (initialization threshold) are fixed as training-optimal values from a separate grid search (as discussed in Section 5.3, the remaining degrees of freedom at inference are the block size $n_{\text{token\_seq\_len}}$ and the $n$-gram verification size, which jointly determine how much parallel draft and verify work is done per step under a given hardware constraints.

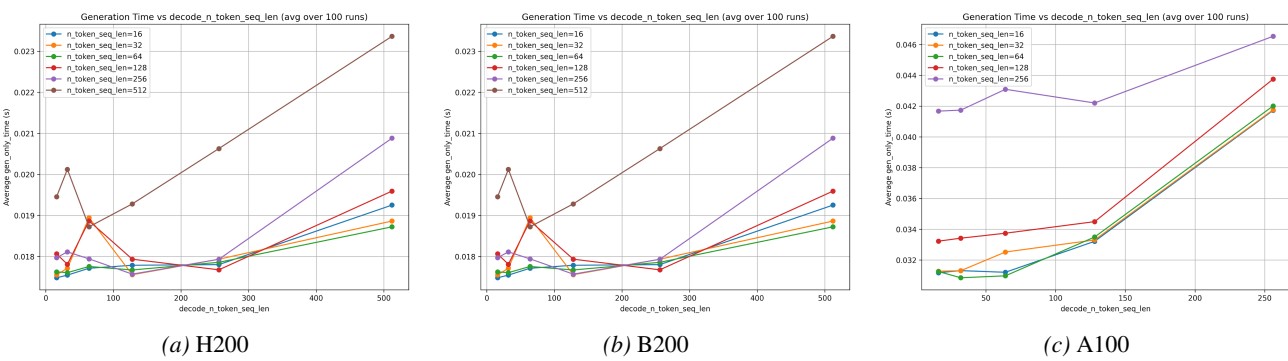

*(a)* H200           *(b)* B200           *(c)* A100

*Figure 7.* Generation-only latency versus total number of parallel decoded tokens across three hardware platforms (A100, H200, B200).

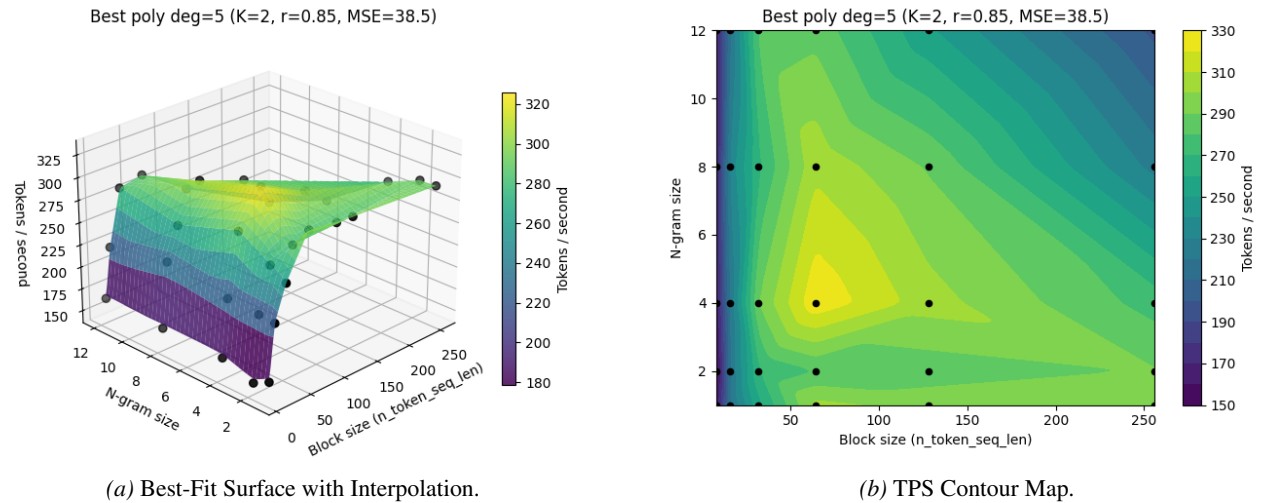

*(a)* Best-Fit Surface with Interpolation.           *(b)* TPS Contour Map.

*Figure 8.* Tokens-per-second (TPS) as a function of block size $n_{\text{token\_seq\_len}}$ and $n$-gram verification size for $K = 2$ and $r = 0.85$. Black dots indicate measured configurations; the surface and contours are obtained by Gaussian-like Smoothing.

To explore this space, we perform a grid search over block sizes $n_{\text{token\_seq\_len}} \in \{8, 16, 32, 64, 128, 256\}$ and $n$-gram verification sizes $n_{\text{gram}} \in \{1, 2, 4, 8, 12\}$, measuring the achieved tokens per second for each pair on the target GPU. Since the raw grid is relatively coarse, we fit a smooth surface over the discrete measurements and use it as a surrogate for continuous hyperparameter selection. Specifically, we construct a 2D polynomial design matrix in (block size, $n$-gram size) of total degree up to 6, select the best degree by mean squared error, and then interpolate the fitted surface onto a dense grid using `scipy.interpolate.griddata` with a light Gaussian-like smoothing pass.

The results are shown in Figure 8, and the resulting surfaces reveal a clear optimum region: tokens-per-second peaks at moderate block sizes and medium $n$-gram verification, with the global maximum near $n_{\text{token\_seq\_len}} \approx 64$ and $n_{\text{gram}} \approx 4$. Very small blocks or $n$-gram verification size underutilize the available FLOPs, while very larger choices push the system closer to the roofline and begin to degrade wall-clock latency. This analysis justifies the final choice of using block size $64$ and $n$-gram size $4$ on B200, which lies near the empirical optimum under each GPU's FLOPs budget.

