# OpenReview forum: "Fast and Accurate Causal Parallel Decoding using Jacobi Forcing"
_ICML.cc/2026/Conference — ICML 2026 regular_

### Official Review · Reviewer_GKQx · 2026-03-10

**Soundness:** 4
**Presentation:** 4
**Significance:** 3
**Originality:** 3
**Overall Recommendation:** 6
**Confidence:** 4

**Summary:**

This paper introduces Jacobi-Forcing LLMs (JF-LLMs) which push the Pareto frontier of generation quality vs latency over ARs, dLLMs, and Speculative Decoding. The work adapts AR-LLMs to generate multiple tokens at a time by enhancing Jacobi Decoding with Jacobi Forcing, a consistency distillation technique at progressively noisier levels across blocks, augmented with an AR loss, and implemented with noise-aware causal attention enhanced with progressive distillation towards larger block sizes.

The authros then propose rejection recycling (n-gram pooling from unconverged blocks) and multi-block decoding (observing that some tokens are predicted independently of preceding noise) to optimize inference.

**Compliance With Llm Reviewing Policy:**

Affirmed.

**Final Justification:**

I maintain my positive view of this paper given the impressive wholistic optimizations and experiments to push the Pareto frontier between generation quality and latency.

**Key Questions For Authors:**

See questions in "weaknesses" section.

**Limitations:**

While there's no overt limitations section, I believe the various trade-offs and optimizations discussed in this paper give a wholistic view of the JF-LLM's strengths and limitations.

**Strengths And Weaknesses:**

## Strengths

This is a strong paper. Jacobi Forcing is a well-motivated proposal for consistency distillation to adapt AR models for parallel decoding, but the main strength comes from the all-round high-quality optimized implementation that showcases the benefits of the approach in the experimental results. This approach presents itself as a strong competitor to speculative decoding without requiring a separate draft model. Experimental results are compelling and up-to-date in their comparisons.

- Multi-block decoding is a very creative heuristic to accelerate decoding speed

## Weaknesses

A number of the heuristics could likely be further explored and optimized, although this doesn't detract from the impact of this paper with their current settings. In particular:
- is a linear progressive noise schedule optimal? While it's better than random or reverse, could other monotonically increasing noise schedules yield better results?
- what is the optimal weight $w$ for the AR loss?
- could rejection recycling be extended with variable-length n-grams?
- an ablation to tease out the individual impacts of rejection recycling versus multi-block decoding would be helpful
- on both HumanEval and MBPP, JF-LLM results in a ~4 point drop in accuracy from AR. What are the fastest settings of JF-LLM that would close the accuracy gap, and what would the speedup look like with those settings? That would make for an important apples-to-apples comparison with the speculative decoding baseline of EAGLE-3, which currently maintains a competitive speedup to JF-LLM while not losing accuracy

---

> ### Author Rebuttal · Authors · 2026-03-31
>
> We thank Reviewer GKQx for recognizing the novelty and effectiveness of Jacobi Forcing. Below, we address each of the concerns point by point:
>
> ### **W1: Is a linear progressive noise schedule optimal? While it's better than random or reverse, could other monotonically increasing noise schedules yield better results?**
>
> To address this, we have added additional experiments using a non-linear monotonically increasing noise schedule (i.e., quadratic progressive). Specifically, we divide the 0%–100% noise range into windows following a quadratic progression. The results of **accuracy and iter/token** on HumanEval are shown in the table below. As observed, the quadratic progressive scheduler is slightly less effective than the linear progressive one.
>
> | window size | linear progressive | quadratic progressive|
> |-------------|--------------------|------------------------|
> | w8          | 84.7 & 0.48        | 82.9 & 0.50            |
> | w16         | 81.7 & 0.46        | 83.0 & 0.49           |
> | w32         | 83.8 & 0.49        | 83.0 & 0.50           |
> | w64         | 83.4 & 0.51        | 83.2 & 0.51           |
>
>
> ### **W2: What is the optimal weight for the AR loss?**
>
> We employ $w=1$ to balance the consistency loss and AR loss in Jacobi Forcing. In practice, this yields the best trade-off between speedup and accuracy. We will clarify this design choice in the revised version.
>
> ### **W3: Could rejection recycling be extended with variable-length n-grams?**
>
> We would like to clarify that our rejection recycling mechanism already inherently supports variable-length n-grams. As described in Section 4.2, the n-gram pool is constructed from noisy token sequences observed at unconverged points during Jacobi decoding. Since the lengths of these noisy token sequences naturally vary depending on the model's trajectory at each step, the retrieved n-grams are not restricted to a fixed length.
>
> ### **W4: An ablation to tease out the individual impacts of rejection recycling versus multi-block decoding would be helpful**
>
> Thanks for the valuable suggestion. We have applied multi-block decoding (MD) and rejection cycling (RC) separately on the math tasks and report the resulting TPF below by fixing the block initialization threshold at $r=0.85, K=2$, and scanning over block size = 8, 16, 32, 64, 128, 256. We report the highest TPF achieved.
>
> |  Strategy | GSM8k | MATH  |
> |---------|------|----------------|
> | neither | 3.62  |  3.66  |
> | MD only   |  3.71  |  3.74  |
> | RC only   | 3.88  |  3.86  |
> | MD+RC   | 4.04 | 3.98   |
>
> As shown, each component provides a clear gain, and combining both yields the best performance. We will include this ablation in the revised version.
>
> Block size and window size choice combinations affect generation quality and speedup in Jacobi Forcing.
> To provide a more quantitative analysis on training settings and speed-accuracy tradeoff, we conducted a broader grid search over block sizes $n= 64, 32$ and varying window sizes $w$, with the same inference configuration used for all runs when measuring TPF, as shown in the table.
>
> | n | window size | 25k steps (Acc / TPF) | 50k steps (Acc / TPF) | 75k steps (Acc / TPF) | 100k steps (Acc / TPF) |
> |---:|:------|:----------------------|:----------------------|:----------------------|:-----------------------|
> | 32 | 32 | 86.0 / 3.17 | 85.4 / 3.42 | 85.4 / 3.46 | 83.5 / 3.62 |
> | 32 | 16 | 86.0 / 3.40 | 86.0 / 3.70 | 85.4 / 3.97 | 84.8 / 4.18 |
> | 32 | 8  | 84.8 / 3.48 | 84.8 / 3.76 | 83.5 / 4.02 | 82.9 / 4.26 |
> | 64 | 64 | 85.4 / 3.21 | 85.4 / 3.29 | 84.8 / 3.46 | 83.5 / 3.57 |
> | 64 | 32 | 85.4 / 3.51 | 84.8 / 3.77 | 83.5 / 4.18 | 83.5 / 4.44 |
> | 64 | 16 | 86.0 / 3.47 | 85.4 / 3.83 | 85.4 / 4.08 | 82.9 / 4.52 |
>
> The results from Table 1 show a trade-off to mitigate performance loss: aggressive settings with larger block size and smaller window size (where more noise allocated to each block in a sequence) can improve TPF but often reduce accuracy, whereas moderate settings achieve a better balance. In particular, $n=32,w=16$ gives one of the best overall accuracy–TPF trade-off, with around 1~2% performance loss only on HumanEval and 3.4x speedup, which is still higher than EAGLE-3 and HASS as reported in Table 1.
>
> We hope these clarifications and additional results fully address your initial concerns and would be happy to discuss any details further!

---

> > ### Author Rebuttal · Reviewer_GKQx · 2026-04-01
> >
> > Thank you for addressing my questions! I maintain my positive view of this work.

---

### Official Review · Reviewer_dNsM · 2026-03-13

**Soundness:** 3
**Presentation:** 2
**Significance:** 3
**Originality:** 2
**Overall Recommendation:** 4
**Confidence:** 4

**Summary:**

This paper proposes Jacobi Forcing (JF), a training and inference framework for accelerating autoregressive (AR) language models via causal parallel decoding, without adopting a full diffusion‑LM (dLLM) formulation. The core idea is to improve upon Consistency LLMs (CLLMs) by introducing a progressive distillation objective, where the model is trained on its own Jacobi decoding trajectories under progressively noisier block contexts, rather than conditioning on fully noisy long contexts. This mitigates the pretrain–posttrain mismatch observed in AR‑adapted dLLMs and improves the model’s ability to predict correct future tokens under noisy context.

Beyond training, the paper introduces rejection recycling and multi‑block decoding at inference time to further boost throughput by reusing high‑quality n‑grams and refining multiple blocks in parallel. These ideas are conceptually related to prior work such as lookahead decoding and Spiffy, but are integrated here within a Jacobi‑based causal framework. Extensive experiments on coding and math benchmarks show that JF‑LLM achieves up to 3.8× speedup with minimal accuracy degradation, and up to ~4× speedup when combined with multi‑block decoding and rejection recycling, outperforming several AR‑based and diffusion‑based baselines.

**Compliance With Llm Reviewing Policy:**

Affirmed.

**Final Justification:**

The authors resolved my concerns for which I've updated my score accordingly

**Key Questions For Authors:**

1. Terminology: token acceptance.
 - The term “accepted token” is used throughout the paper. Can the authors explicitly clarify that acceptance refers to correctness under speculative / Jacobi verification (as in speculative decoding literature), and distinguish it from intermediate or pseudo‑accepted tokens?

2. Why are dLLMs slower than AR models in practice?
 - Figure 1 suggests that dLLMs have lower throughput than AR models. In the worst case, dLLMs could unmask one token per step. Is this gap primarily due to the lack of optimized bidirectional attention kernels (e.g., FlashAttention support), or due to decoding strategy choices? What unmasking strategy is used for dLLM baselines (single‑token vs. multi‑token)? Since TPS can vary significantly, this detail seems important.


3. Efficiency claims in related work.
 - The related‑work section notes that dLLMs are less efficient than AR models while acknowledging that they perform multi‑token prediction. Can the authors elaborate on this apparent contradiction?


4. Missing and incomplete related work (minor).
 - A vocabulary‑trimming approach such as VOCABTRIM [1] is missing from the speculative decoding discussion, and there appears to be a typo in the Jacobi decoding paragraph. Could these issues be addressed?


5. Complexity reduction via progressive noise.
 - The paper claims that the longest noisy span reduces from $O(nN)$ to $O(⌈tn⌉)$. Can the authors provide a more intuitive explanation of why conditioning on progressively noisy blocks achieves this reduction?


6. Sequence packing vs. BiTA [2].
 - Sequence packing and interleaved attention masking are also discussed in BiTA. Can the authors clarify what is technically different here, or explicitly acknowledge the overlap?


7. Training on target‑model trajectories.
 - Using trajectories generated by the target model for training is also discussed in prior work on draft‑model alignment [3]. It would be nice to differentiate current approach from past work.


8. Novelty of rejection recycling and multi‑block decoding.
 - Rejection recycling resembles ideas in lookahead decoding, and multi‑block decoding is conceptually related to Spiffy [4]. How much of the final speedup comes from these inference techniques versus the proposed progressive distillation training?

9. Figure clarity.
 - In Figure 4, the meaning of the red tokens is unclear. Would a text‑level (rather than token‑ID‑level) visualization make the intuition clearer?

10. Window‑size sensitivity.
 - In Table 4, why does accuracy drop by approximately 3 points for window size 16 compared to 8 or 32? Is this due to optimization instability or over‑regularization?


11. Full noise‑conditioned mask vs. learning signal.
 - Table 5 suggests that a fully noise‑conditioned mask improves training, yet earlier the paper argues that long noisy contexts provide poor learning signals. Can the authors reconcile these observations?


12. Attributing gains to training vs. inference.
- What happens if rejection recycling and multi‑block decoding are applied to vanilla CLLM? This would help isolate the gains from the progressive distillation objective itself.

References

[1] Goel, Raghavv, et al. "Vocabtrim: Vocabulary pruning for efficient speculative decoding in llms." arXiv preprint arXiv:2506.22694 (2025).

[2] Lin, Feng, et al. "Bita: Bi-directional tuning for lossless acceleration in large language models." Expert Systems with Applications 279 (2025): 127305.

[3] Goel, Raghavv, et al. "Direct alignment of draft model for speculative decoding with chat-fine-tuned llms." arXiv preprint arXiv:2403.00858 (2024).

[4] Agrawal, Sudhanshu, et al. "Spiffy: Multiplying diffusion llm acceleration via lossless speculative decoding." arXiv preprint arXiv:2509.18085 (2025).

**Limitations:**

While Jacobi Forcing shows strong empirical performance, the final speedups rely on a combination of training‑time and inference‑time techniques whose individual contributions are not fully disentangled. Several comparisons depend on system‑level choices (attention kernels, unmasking strategies) that may vary across implementations.

Other weaknesses/questions are mentioned above

**Strengths And Weaknesses:**

# Strengths

- Addresses a key limitation of AR‑adapted dLLMs: The paper correctly identifies the pretrain–posttrain mismatch introduced by bidirectional attention and fully masked contexts in dLLMs, and proposes a causal alternative that better aligns with AR pretraining.

 - Well‑motivated progressive distillation objective: Progressive noise scheduling over blocks is a principled improvement over prior CLLM objectives that condition on fully noisy long contexts, and is supported by both theoretical intuition and ablations.

- Strong systems‑level performance: The combination of Jacobi Forcing with rejection recycling and multi‑block decoding delivers substantial wall‑clock speedups on modern GPUs, with careful analysis of FLOPs utilization and roofline effects.

- Broad and competitive empirical evaluation: The paper compares against a wide range of baselines, including AR decoding, CLLM, speculative decoding (EAGLE‑3, HASS), and multiple dLLMs, demonstrating favorable speed–quality trade‑offs.

# Weaknesses

- Novelty relative to prior inference optimizations is not fully disentangled: Several inference‑time techniques (rejection recycling, multi‑block decoding, sequence packing) closely resemble ideas in lookahead decoding, Spiffy, and BiTA, but the paper does not clearly isolate the gains attributable specifically to the progressive distillation objective.

- Terminology and exposition issues reduce clarity: Key terms such as “token acceptance” are used loosely, and some figures and equations lack sufficient explanation or contain typos, making the paper harder to follow.

- Throughput comparisons raise fairness questions: Comparisons between AR models and dLLMs may be influenced by differences in attention implementations (causal vs. bidirectional) and unmasking strategies, which are not always made explicit.

- Some empirical behaviors are insufficiently explained: Accuracy drops for certain window sizes and the behavior of fully noise‑conditioned blocks vs standard noisy context is not differentiated properly.

---

> ### Author Rebuttal · Authors · 2026-03-31
>
> We thank reviewer dNsM for the constructive feedback. We address each of your concerns point by point:
>
> ### **W1, Q12: The paper does not clearly isolate the gains attributable specifically to the progressive distillation objective. What happens if rejection recycling and multi‑block decoding are applied to vanilla CLLM?**
>
> We thank the reviewer for bringing up the related work. We want to clarify inference optimizations introduced in this paper only overlap with them at either implementation-level but for different purposes (Spiffy, BiTA, see our responses to Q6, Q8) or at motivation level but with different implementation and models (lookahead decoding, see our response to Q8).
>
> To address the question on attributing gains to training vs. inference, we add an additional experiment to test both JF-LLM and CLLM with and without inference optimizations. The results show even without the inference-time techniques, our models trained after progressive distillation achieve 3.50x speedup on gsm8k, which significantly outperforms rejection recycling and multi‑block decoding applied to CLLM (2.15x).
>
> |  Strategy | gsm8k | MATH  |
> |-|-|-|
> | vanilla CLLM (w/o MR)   |  2.08  |  2.04  |
> | vanilla CLLM (w/ MR)   | 2.15  |  2.06  |
> | JF-LLM (w/o MR) | 3.50  |  3.65  |
> | JF-LLM (w/ MR)   | 3.71 | 3.68   |
>
> ### **W1, Q6: Sequence packing vs. BiTA [2]**
>
> BiTA and Jacobi Forcing use sequence packing for different purposes. BiTA (1) adds bidirectional tuning for prompt and mask tokens, and (2) uses inference-time tree flattening with custom attention for joint generation and verification in one pass. In contrast, Jacobi Forcing does not use tree decoding, its sequence packing is meant to enable noise-conditioned generation and serve as an efficiency mechanism to compute distillation and AR losses efficiently in one pass.
>
> ### **W1, Q8: Novelty of rejection recycling (vs. lookahead decoding) and multi‑block decoding (vs. Spiffy). How much of the final speedup comes from these inference techniques versus the proposed progressive distillation training?**
>
> Rejection recycling shares the high-level goal of verifying extra n-gram candidates to accelerate causal parallel decoding, but differs substantially in design and FLOPs cost. Rejection recycling uses simple prefix-suffix matching to extract candidates and verifies them in the batch dimension with a shared KV cache, avoiding the complex attention modifications required by lookahead decoding. It is also effective with a small n-gram pool (e.g. 4 in practice), thereby avoiding the large FLOPs and memory overhead of generating and storing many n-grams.
>
> Spiffy is designed for diffusion LLMs, using a directed draft graph and bidirectional block-wise generation. Whereas JF-LLM multi-block decoding targets AR models and decodes tokens concurrently across multiple blocks.
>
> Regarding the gain from inference techniques, the delta between JF-LLM and JF-LLM (MR) in our manuscript directly reflects the gains from our inference optimizations. Moreover, we ablate the effect of each component in the following table with all the other settings kept the same as Table 1:
>
> |  Strategy | gsm8k | MATH  |
> |-|-|-|
> | neither | 3.62  |  3.66  |
> | MD only   |  3.71  |  3.74  |
> | RC only   | 3.88  |  3.86  |
> | MD+RC   | 4.04 | 3.98   |
>
> The results show the most significant gain is from our progressive distillation objective and combining both MD+RC yields the best performance.
>
> ### **W4, Q10: Window‑size sensitivity. Accuracy drops for certain window sizes and the behavior of fully noise‑conditioned blocks vs standard noisy context is not differentiated properly.**
>
> We have provided an ablation in Table 5 has showed a noise-conditioned mask (method used throughout the paper) provides a significantly better learning signal than the alternative design choice, where blocks from prior window conditioning on clean tokens instead of noisy tokens.
>
> To address the reviewer’s question, we add additional experiments using varying window-size settings and non-linear noise schedules, where a 0%–100% noise range is divided into windows following different progression patterns. The results of **accuracy and iter/token** on HumanEval are shown below. As observed, the quadratic progressive scheduler is slightly less effective than the linear progressive one.
>
> | window size| linear | quadratic | random |
> |-|-|-|-|
> | 8 | 84.7 & 0.48 | 82.9 & 0.50  | 82.3 & 0.48 |
> | 16  | 81.7 & 0.46   | 83.0 & 0.49  | 82.0 & 0.47 |
> | 32  | 83.8 & 0.49   | 83.0 & 0.50  | 83.5 & 0.49 |
> | 64  | 83.4 & 0.51   | 83.2 & 0.51  | 83.4 & 0.49 |
>
> Due to the word limit, we provide responses to the remaining clarification questions and suggestions on an anonymous page: [https://pagedrop.dev/s/sttD2zNX/](https://pagedrop.dev/s/sttD2zNX). We are happy to further discuss during the discussion period. Given the clarifications and additional experiments in response to your initial concerns, we hope for your reconsideration in raising your score.

---

> > ### Author Rebuttal · Reviewer_dNsM · 2026-04-04
> >
> > Thanks for the responses, unfortunately I'm unable to access the link provided by the authors.
> > Could they please provide the rest of the responses here

---

> > > ### Author Response · Authors · 2026-04-04
> > >
> > > ### **W2, Q1: terminology clarification and distinguishing it from intermediate or pseudo‑accepted tokens**
> > >
> > > We thank the reviewer for the suggestion to improve the clarity of our terminology. Jacobi decoding introduced in the preliminaries serves as the foundation of our work, we use "acceptance" and "accepted tokens" in accordance with prior Jacobi decoding formulations, defined as tokens accepted from left to right with rejection sampling in each Jacobi iteration, including both greedy and non-greedy settings. We will make it clearer in the revision.
> > >
> > > ### **W3: comparison fairness due to dLLM bidirectional attention.**
> > >
> > > Our experiments intend to compare a range of parallel decoding methods. Depending on the method, this can involve different attention implementations. However, all models are trained from the same base model (Tables 1 and 3), using the same training data, and same system settings (Table 2). So we believe the comparisons are fair. If the reviewer has additional concerns, we are happy to discuss further.
> > >
> > > ### **Q2, Q3: Why are dLLMs slower than AR models in practice (attention kernel or unmasking strategy reason)? why acknowledging that dLLMs perform multi‑token prediction but they are slower?**
> > >
> > > All dLLMs baselines we run use multi-token decoding. By slower we mean dLLMs' TPF could be higher, but for some **end-to-end latency could still be higher (TPS is lower)** This is because each forward pass for dLLM involves more FLOPs, and it scales with the block size. **TPF needs to be sufficiently high in order to compensate this overhead**. You can find more discussion on TPF vs. FLOPs tradeoff for end-to-end latency optimization in Appendix D.
> > >
> > > Moreover, unlike AR models where each generated token will be committed to KV cache, efficient KV cache reuse is lacking in many dLLM models since the generation is in any order and the attention mask is bidirectional, resulting in an unconverged KV until the entire block has completed decoding.
> > >
> > > ### **Q4: related work and typo.**
> > >
> > > Thanks for pointing it out. We will add VOCABTRIM to the related work in our revision and fix the “text-to-image” typo.
> > >
> > > ### **Q5: Complexity reduction via progressive noise.**
> > >
> > > Consider the “worse-case scenario” using a random schedule and all blocks are assigned with a max noise level, for all $N$ blocks, each has $n$ noisy tokens, then the longest noisy token span would be $nN$ tokens. On the other hand, with a progressive noise schedule, the longest noisy token span will be determined by the block with highest noise ratio $t$, therefore reducing noisy token span to at most $tn$ tokens.
> > >
> > > ### **Q7: differentiate the current approach from past work that use trajectories generated by the target model for draft‑model alignment.**
> > >
> > > In TVD++ and OSD, the goal is to train or adapt a separate small draft model for speculative decoding so that it better matches a fixed target model, using target-provided responses or corrections together with distillation-based objectives. However, Jacobi Forcing does not train a separate drafter; instead, it trains the AR model itself into a parallel decoder. We will clarify this distinction in the related work section.
> > >
> > > ### **Q9: Figure 4 clarity.**
> > >
> > > In Figure 4, the tokens highlighted in red in iteration 3 denote consecutive correct tokens that exactly match those in the final (fixed-point) iteration (also highlighted in red), they are simply misplaced, as noted in the caption.
> > >
> > > We agree that a text-level visualization would make the intuition clearer. We will revise the figure and caption accordingly to make the meaning of the red tokens more explicit.
> > >
> > > ### **W4, Q10: Window‑size sensitivity.**
> > >
> > > To provide a more comprehensive overview on the effects of different window size choices, we added additional experiments with 3 additional runs and taken 3-run average for the different settings. The results of accuracy and iter/token on HumanEval are shown in the table below. As observed, the generation quality results are more statistically stable and exhibit a trend where **smaller window size choice provides faster speedup**.
> > >
> > > |window size|linear progressive|random|
> > > |:-:|:-:|:-:|
> > > |8|84.1 & 0.46|83.1 & 0.52|
> > > |16|83.5 & 0.47| 83.4 & 0.52|
> > > |32|83.7 & 0.49|83.2 & 0.53|
> > >
> > > ### **Q11: Table 5 ablation learning signal.**
> > >
> > > We would like to clarify that the noise-conditioned mask (NC) entry refers to applying the complete noise-conditioned mask shown in Figure 2. In this setting, the longest noisy token span is $O(tn)$, as explained in our response to Q5. This is different from applying the maximum noise ratio $t=1$ to all blocks (where the result is worse since it provides poor learning signal), which would produce the longest possible noisy context span. Therefore, these observations are not contradictory. We will make this interpretation clearer in the revision.
> > >
> > > We welcome further discussion and kindly ask for reconsideration in raising your score if these clarifications resolve your concerns.

---

### Official Review · Reviewer_QCHW · 2026-03-13

**Soundness:** 2
**Presentation:** 3
**Significance:** 2
**Originality:** 3
**Overall Recommendation:** 4
**Confidence:** 4

**Summary:**

This paper proposes a progressive distillation technique for training LLMs for efficient parallel decoding. Specifically, the authors utilize Jacobi forcing to train a multi-token predictor to converge faster along Jacobi decoding trajectories. Empirically, the trained model achieves greater efficiency and lower latency.

**Compliance With Llm Reviewing Policy:**

Affirmed.

**Final Justification:**

Thank you to the authors for their engaged discussion and for further addressing my concerns about pure sampling and distribution mismatch. I will increase my score, and I recommend that the authors include the pure sampling results on corresponding benchmarks in the revised version.

**Key Questions For Authors:**

1. In the abstract and introduction, the paper mentions the issue of "pretrain-to-posttrain" distribution mismatch. It is not clear how the proposed method solves this issue, except for Equation 9. Most components in Section 4 do not seem to be related to mitigating this mismatch. Could the authors clarify whether mitigating this issue is a primary goal or just a motivation?

2. According to Jacobi decoding in Section 3.1, this method seems to only work on greedy decoding. Could the author clarify how it applies to a pure sampling setting?

3. Can the author explain why there is a huge accuracy drop in coding benchmarks, as shown in Table 1? Could the authors elaborate more on this speed-accuracy tradeoff and explain why the proposed approach is still preferable in these settings?

**Limitations:**

yes

**Strengths And Weaknesses:**

## Soundness
- The submission is technically sound, and the claims are supported by experimental results. The experimentals are well-designed and cover multiple relevant benchmarks.
- In the abstract and introduction, the paper mentions the issue of "pretrain-to-posttrain" distribution mismatch. It is not clear how the proposed method solves this issue, except for Equation 9. Most components in Section 4 do not seem to be related to mitigating this mismatch. Could the authors clarify whether mitigating this issue is a primary goal or just a motivation?


## Presentation
- The general presentation is good and easy to follow. The main method is described clearly, and experiments are rigorous.
- However, this paper seems to be an improvement over the CLLM (Kou et al. 2024) paper, but this prior work is not sufficiently introduced. I suggest that the authors add a section before section 4.1 to briefly discuss CLLM and its limitations, as well as highlight their specific distinctions from CLLM.


## Significance
- The inference efficiency is an important problem, and this paper addresses this by training a multi-token predictor for parallel decoding. The scope of impact is broad.
- However, empirically, there is a significant accuracy drop as shown in Table 1. However, previous methods like CLLM and speculative decoding maintain the same accuracy.


## Originality

- The paper proposes to utilize Jacobi forcing to mitigate the current bottleneck of efficient decoding and provide practical insights.

---

> ### Author Rebuttal · Authors · 2026-03-31
>
> We thank reviewer QCHW for this insightful questions. We address each of your question and concern point by point below:
>
> ### **Q1, W1 (soundness): the paper mentions the issue of "pretrain-to-posttrain" distribution mismatch. It is not clear how the proposed method solves this issue.**
>
> When we mention “pretrain-to-posttrain”, we are referring to the significant discrepancy between artificially masked data distribution using bidirectional attention in dLLM post-training and real natural language data distribution using causal attention for AR model pretraining. Jacobi Forcing addresses this gap by preserving the causal attention and training the model on its own generated Jacobi trajectories. We will add this clarification in the revised version.
>
> ### **Q2: According to Jacobi decoding in Section 3.1, this method seems to only work on greedy decoding. Could the author clarify how it applies to a pure sampling setting?**
>
> In a non-greedy sampling setting, Jacobi decoding faces a limitation: the randomness in stochastic equations prevents us from leveraging a deterministic fixed point with full reproducibility during Jacobi iterations. Consequently, our current work primarily focuses on greedy decoding.
>
> However, we provide some initial insights into addressing this limitation. By leveraging the sequential nature of LLMs, we can modify the Jacobi iteration to support stochastic techniques (e.g., top-k or top-p sampling) while ensuring fixed-point convergence. Specifically, for a randomly initialized n-token sequence $y^{(0)} = \\left\\{ y_1^{(0)}, \\dots, y_n^{(0)} \\right\\}$, we can update it following the pseudocode below and set the output as the fixed point:
>
> ```python
> Input: LLM p(x), randomly initialized n-token sequence y^(0)
> Output: sampled fixed point y*
>
> for i = 1 to n do
>     for j = 1 to n do
>         if j < i then
>             y_j^(i) = y_j^(i-1)
>         else
>             y_j^(i) = Stochastic_Sampling(p(y | y_<j^(i), x))
>     if y^(i) = y^(i-1) then
>         break
> return y^(i)
> ```
>
> We will further explore this idea and conduct experiments to verify its effectiveness in the context of stochastic sampling strategies.
>
>
> ### **Q3, W3 (significance): Can the author explain why there is a huge accuracy drop in coding benchmarks, as shown in Table 1? Could the authors elaborate more on this speed-accuracy tradeoff and explain why the proposed approach is still preferable in these settings?**
>
> Adopting a more aggressive block size and noise schedule during training naturally introduces a trade-off: it yields significantly higher speedups at the cost of minor performance degradation. We believe this trade-off is favorable and provides the community with an additional design point when lower latency is preferred. While there is a modest accuracy drop of 1~4% relative to the baseline (depending on tasks and configurations), JF-LLM achieves substantial throughput gains. Specifically, it delivers a 58% speedup over CLLM (163.9 vs. 103.3 TPS) and a 28% speedup over very competitive speculative decoding baseline, EAGLE-3 (163.9 vs. 127.2 TPS).
>
>
> ### **W2 (presentation): this paper seems to be an improvement over the CLLM paper, but this prior work is not sufficiently introduced. I suggest that the authors add a section before section 4.1 to briefly discuss CLLM and its limitations, as well as highlight their specific distinctions from CLLM.**
>
> We thank the reviewer for making this connection with CLLM and the suggestion. We want to kindly highlight that we already have section 3.2 about the consistency distillation technique adopted by CLLM as well as dParallel. In the introduction, we have also mentioned CLLM’s major limitation: as block size increases for CLLM, the number of tokens correctly decoded per iteration remains essentially constant for many existing techniques. We will integrate the reviewer's suggestion and make further clarification on the distinction with CLLM in our revision.
>
> To make the distinction more clear, we add additional experiment results to compare JF-LLM with CLLM at different block sizes, and report the **TPS speedup over AR** in the table below (all exp settings are the same as Table 1):
>
> | Method (at different block sizes) | 16 |   32     |    64     |    128     |    256     |
> |------------|-------------------|------------|------------|-------------|-------------|
> | CLLM |  $2.4\times$    | $2.5\times$ | $2.5\times$ | $2.4\times$ | $2.0\times$ | 87.8 |
> | JF-LLM | $2.7\times$ | $3.5\times$ | $\mathbf{3.9\times}$ | $3.7\times$ | $3.3\times$ |
>
> we can see JF-LLM consistently provides higher speedups than CLLM at the same block size, especially at $k=32, 64$.
>
> Given the clarifications on our method and additional experiments in response to your initial questions, we hope for your reconsideration in raising your score if our rebuttal has addressed your concern. We would also be more than happy to discuss any remaining questions or concerns you may have.

---

> > ### Author Rebuttal · Reviewer_QCHW · 2026-04-04
> >
> > Thank you to the authors for their detailed response and additional experiments.
> > My concerns regarding Q1 and Q2 remain.
> >
> > > When we mention “pretrain-to-posttrain”, we are referring to the significant discrepancy between artificially masked data distribution using bidirectional attention in dLLM post-training and real natural language data distribution using causal attention for AR model pretraining.
> >
> > Q1. I am a bit confused by the use of the term "distribution" here. My understanding is that the data distribution refers to the underlying data itself, whereas bidirectional vs. causal attention is a modeling or architectural choice. These seem to be two different aspects. Besides, I believe the “masked data distribution” describes the corruption process in the diffusion training objective, rather than a true data distribution.
> >
> > > In a non-greedy sampling setting, Jacobi decoding faces a limitation: xxx
> >
> > I think this is one of the limitations of this method and hinders its applicability.
> >
> > Thus, I prefer to maintain my score.

---

> > > ### Author Response · Authors · 2026-04-05
> > >
> > > ### **Q1: Confusion around the term ‘distribution’.**
> > >
> > > We appreciate the reviewer’s careful reading and apologize for the confusion around our wording. We acknowledge the data corpus (underlying data) used is the same for both AR models and dLLMs. Meanwhile, we want to clarify that when referring to the “pretrain-to-posttrain” mismatch, we mean a mismatch in the **training setup** between pre-training and post-training, including: (1) how the training inputs are constructed (unmasked vs. masked), (2) attention implementation (causal vs. bidirectional). Jacobi Forcing addresses this gap by keeping both the same as how AR model pre-training is conducted with the causal attention preserved and training the model on its own generated Jacobi trajectories. We will add this clarification in the revised version.
> > >
> > >
> > >
> > > ### **Q2: Non-greedy sampling setting.**
> > >
> > > Thanks for raising this follow-up question. Our original rebuttal was meant to highlight that the randomness in stochastic equations **prevents us from leveraging deterministic fixed points with full reproducibility** for training and evaluation as reported in e.g. Table 1 and Table 3.
> > >
> > > Besides of reproducibility, we would like to firmly clarify that **there is no fundamental limitation preventing JF-LLM from supporting non-greedy decoding**. We have actually already implemented a parallel decoding algorithm for non-greedy decoding, **as provided in our supplementary materials** ‘modeling/cllm2_qwen2_modeling_two_condition.py’ (with top-k and top-p sampling support), and the implementation incorporates rejection sampling for verification. Specifically, it leverages the logits of unaccepted draft tokens to form the new draft distribution, acting as a self-speculative decoding mechanism without requiring an extra draft model.
> > >
> > > To further address your concern on non-greedy decoding applicability, we provide additional experiment results on HumanEval with our non-greedy (top-p=0.9 and top-k=20) parallel decoding algorithm as follows. As seen, our JF-LLM also generalizes well in the non-greedy setting with more than 3x end-to-end speedup using vanilla Jacobi decoding.
> > >
> > > | Models | Acc | TPS | TPF |
> > > |:--:|:--:|:--:|:--:|
> > > | Original (non-greedy) | 84.7  | 38.9 | 1.0 |
> > > | JF-LLM (non-greedy) | 80.3  |  120.9 | 3.3 |
> > >
> > > If the reviewer has any remaining concerns, we would be more than happy to discuss further and make any further clarifications!

---

### Official Review · Reviewer_oZBG · 2026-03-13

**Soundness:** 3
**Presentation:** 3
**Significance:** 3
**Originality:** 3
**Overall Recommendation:** 4
**Confidence:** 3

**Summary:**

This paper proposes Jacobi Forcing, a training framework for enabling efficient causal parallel decoding in autoregressive large language models. The work aims to address the mismatch between autoregressive pretraining and diffusion-style post-training used in recent diffusion-based language models. Instead of adapting AR models into diffusion-style architectures, the authors directly train AR models to function as parallel decoders through a progressive consistency distillation strategy.

The proposed approach introduces a progressive noise schedule, a noise-aware causal attention mechanism, and a sequence packing technique to improve training efficiency. Additionally, the authors propose two inference optimizations (rejection recycling and multi-block decoding) to further increase token acceptance per iteration. Experimental results on coding benchmarks (HumanEval, MBPP) and mathematical reasoning benchmarks (GSM8K, MATH) demonstrate up to 3.8× generation speedup while largely maintaining generation quality compared to standard autoregressive decoding.

**Compliance With Llm Reviewing Policy:**

Affirmed.

**Final Justification:**

According to the rebuttal, I would keep my evaluation.

**Key Questions For Authors:**

1. Can the authors provide a training cost comparison between Jacobi Forcing and alternative approaches such as speculative decoding with draft models?
2. The paper reports slight accuracy degradation on certain benchmarks. Are there training configurations or hyperparameters that mitigate this trade-off?
3. How sensitive is the method to block size and noise schedule design in larger-scale settings?

**Limitations:**

1. The proposed system introduces multiple inference optimizations, making it unclear which components contribute most to the overall gains.
2. The paper does not provide a detailed training compute comparison against competing approaches.

**Strengths And Weaknesses:**

### Strengths

1. The Jacobi Forcing method introduces a progressive noise schedule and noise-aware causal masking that allow models to learn parallel token prediction under noisy contexts. This is a meaningful extension over previous approaches such as consistency distillation in CLLM.
2. The linear progressive noise schedule, noise-aware causal attention, and sequence packing technique are innovative and effective.
3. Rejection recycling and multi-block decoding are hardware-aware and achieve meaningful speedup gains without additional model training.
4. Comprehensive and rigorous experimental validation.

### Weaknesses

1. The experiments are limited to coding and math tasks with 7B-scale LLMs. There is no evaluation on natural language generation tassks, longer sequence generation, or larger model scales, which limits the generalizability of the results.
2. While the paper shows JF-LLM generates high-quality draft tokens under noisy context, it does not provide a qualitative or quantitative analysis of how the model learns to denoise noisy tokens.
3. Lacking of analysis on training cost and efficiency.

---

> ### Author Rebuttal · Authors · 2026-03-31
>
> We thank reviewer oZBG for the constructive feedbacks and we address each of your question and concern below:
>
> ### **W1: There is no evaluation on natural language generation tasks, longer sequence generation, or larger model scales.**
>
> We added experiments on conversational tasks with longer generations. Specifically, we initialize from Qwen2.5-Coder-7B-Instruct, train on 100k examples from the chat split of Nemotron-Post-Training-Dataset-v2 with sequences up to 4k tokens, and apply Jacobi Forcing using the same settings as for the math and coding tasks. In the table, the numbers in parentheses denote the accuracy change relative to the original AR model, and TPF denotes tokens per forward pass. The results show that our conclusions generalize beyond math and coding to conversational language tasks as well.
>
> | Dataset | TPF $\uparrow$ | Quality Metric $\uparrow$   |
> |---------------|--------------------------|------------------|
> | AlpacaEval     |        2.15           |  45.3 (-1.3)   |
> | MT-Bench  |         2.44            |  8.8 (-0.1)     |
>
> Our experiments are conducted on 7B-scale models due to compute constraints as an academic research team. However, please also note progressive consistency distillation does not rely on size-specific architectural tricks, so we expect the method will transfer to larger models.
>
> ### **W2: Qualitative or quantitative analysis of how the model learns to denoise noisy tokens**
>
> We do have a qualitative analysis shown in figure 4, where you can see the model learns to generate correct long n-gram (commonly used natural language phrases like Python clauses) even when conditioning on noisy tokens. Quantitatively, you can find results in Table 1 and Table 3, where you can see this is not the case when applying Jacobi to pre-trained AR models, and the resulting TPF is marginally better than 1. Whereas for JF-LLM TPF is greater than 4.5.
>
> ### **W3, Q1, L2: Lack of analysis on training cost and efficiency.**
>
> We want to clarify we do have a comparison on training cost with other methods on Figure 1. For clarity, we attach a table with numerical values here for clear comparison, which shows Jacobi Forcing train on around 1B tokens but is able to achieve significant speedup while maintaining good performance on HumanEval.
>
> | Technique | Throughput (TPS) | Pass@1 | Train Tokens (B) |
> |---|---:|---:|---:|
> | qwen-2.5-coder-7b-instruct | 41.3 | 87.8 | 0 |
> | llada-7b-instruct | 20.0 | 45.1 | 0 |
> | dream-7b-instruct | 3.0 | 63.4 | 580 |
> | dream-coder-7b-instruct | 5.0 | 82.9 | 322 |
> | fast-dllm-v2 | 100.0 | 63.4 | 1 |
> | SDAR | 21.5 | 78.7 | 50 |
> | Jacobi Forcing (ours) | 163.9 | 83.5 | 1 |
>
> ### **Q3: How sensitive is the method to block size and noise schedule design in larger-scale settings?**
>
> We have already included ablation on noise schedule in Table 4, which shows linear progressive noise schedule leads to the best learning signal with higher TPS while preserving quality.
>
> To provide a more quantitative analysis on block size choices, we conducted a broader grid search over block sizes $n= 64, 32$ and varying window sizes $w$, with the same inference configuration used for all runs when measuring TPF, as shown in the table.
>
> | n | window size | 25k steps (Acc / TPF) | 50k steps (Acc / TPF) | 75k steps (Acc / TPF) | 100k steps (Acc / TPF) |
> |---:|:------|:----------------------|:----------------------|:----------------------|:-----------------------|
> | 32 | 32 | 86.0 / 3.17 | 85.4 / 3.42 | 85.4 / 3.46 | 83.5 / 3.62 |
> | 32 | 16 | 86.0 / 3.40 | 86.0 / 3.70 | 85.4 / 3.97 | 84.8 / 4.18 |
> | 32 | 8  | 84.8 / 3.48 | 84.8 / 3.76 | 83.5 / 4.02 | 82.9 / 4.26 |
> | 64 | 64 | 85.4 / 3.21 | 85.4 / 3.29 | 84.8 / 3.46 | 83.5 / 3.57 |
> | 64 | 32 | 85.4 / 3.51 | 84.8 / 3.77 | 83.5 / 4.18 | 83.5 / 4.44 |
> | 64 | 16 | 86.0 / 3.47 | 85.4 / 3.83 | 85.4 / 4.08 | 82.9 / 4.52 |
>
> Table 1: ablation study on training hyperparameter choices.
>
> From the results, for all block size choices, we can see as training steps scale, maximum TPS will saturate among correctly generated samples. For larger block size, speedup is more significant at the cost of a bit more performance drop. And adding noise too aggressively (use small window size with the respect to block size) will result in training instability.
>
>
> ### **Q2: The paper reports slight accuracy degradation on certain benchmarks. Are there training configurations or hyperparameters that mitigate this trade-off?**
>
> Accuracy–speed trade-off can be mitigated through the choice of block size $n$ and window size $w$ for any given noise schedule. In particular, the results from Table 1 show a trade-off: aggressive settings with larger block size and smaller window size (where more noise allocated to each block in a sequence) can improve TPF but often reduce accuracy, whereas moderate settings achieve a better balance. In particular, $n=32,w=16$ gives one of the best overall accuracy–TPF trade-off.

---

> > ### Author Rebuttal · Reviewer_oZBG · 2026-04-02
> >
> > Thank the authors for the clarification. My concerns have been mostly addressed with respect to the current status of the study and the resource constraints. I believe the contribution is moderate, and I would keep my positive evaluation.

---

### Decision · Program_Chairs · 2026-04-30

**Decision:**

Accept (regular)

**Comment:**

The paper proposes a Jacobi Forcing, a progressive consistency distillation method that directly trains autoregressive LLMs for causal parallel decoding. Proposed solution achives up to 3.8x wall-clock speedup on coding and math benchmarks.

Reviewers rated soundness good-to-excellent, presentation fair-to-excellent, significance fair-to-good, and originality fair-to-good, reaching consensus on weak accept to strong accept (three weak accepts, one strong accept).

As main strength, reviewers mentioned strong empirical speedups, many ablations, adrresing the pretrain-to-posttrain mismatch of prior dLLMs like fast-dLLM.

As key weaknesses, reviewers initially questioned limited evaluation on natural-language and longer-sequence tasks, insufficient isolation of training vs inference gains, and some novelty overlap with prior techniques such as CLLM.

Authors provided a detailed rebuttal with new experiments on conversational tasks, non-greedy sampling, block-size/noise-schedule ablations, latency comparisons to monolithic models, grouped loss ablations, and explicit distinctions from related methods.

All reviewers who responded acknowledged the rebuttal as resolved or partially resolved, with several explicitly updating their scores positively after the additional results.

The paper is technically sound with well-supported claims, strong empirical validation across benchmarks and settings, and effective rebuttal addressing all major concerns, making its significance and originality sufficient to meet ICML acceptance criteria.